# Preference Discerning with LLM-Enhanced Generative Retrieval

**Fabian Paischer**
*ELLIS Unit, LIT AI Lab, Institute for Machine Learning, JKU Linz, Austria*
*AI at Meta*

**Liu Yang**
*University of Wisconsin-Madison*
*AI at Meta*

**Linfeng Liu, Shuai Shao, Kaveh Hassani, Jiacheng Li, Ricky Chen, Zhang Gabriel Li, Xiaoli Gao, Wei Shao, Xue Feng, Nima Noorshams, Sem Park, Bo Long, Hamid Eghbalzadeh[†]**
*heghbalz@meta.com*
*AI at Meta*

**Reviewed on OpenReview:** *https://openreview.net/forum?id=74mrOdhvvT*

## Abstract

In sequential recommendation, models recommend items based on user's interaction history. To this end, current models usually incorporate information such as item descriptions and user intent or preferences. User preferences are usually not explicitly given in open-source datasets, and thus need to be approximated, for example via large language models (LLMs). Current approaches leverage approximated user preferences only during training and rely solely on the past interaction history for recommendations, limiting their ability to dynamically adapt to changing preferences, potentially reinforcing echo chambers. To address this issue, we propose a new paradigm, namely *preference discerning*, which explicitly conditions a generative recommendation model on user preferences in natural language within its context. To evaluate preference discerning, we introduce a novel benchmark that provides a holistic evaluation across various scenarios, including preference steering and sentiment following. Upon evaluating current state-of-the-art methods on our benchmark, we discover that their ability to dynamically adapt to evolving user preferences is limited. To address this, we propose a new method named Mender (**M**ultimodal Prefer**en**ce **D**iscern**er**), which achieves state-of-the-art performance in our benchmark. Our results show that Mender effectively adapts its recommendation guided by human preferences, even if not observed during training, paving the way toward more flexible recommendation models. [1]

## 1 Introduction

Traditional sequential recommendation refers to the task of recommending items to users based on their historical interactions. This requires inferring latent variables, such as user preferences and intents, which are often not explicitly provided in common datasets, as information about users is usually scarce. Therefore, current works use LLMs to approximate user preferences from the users' interaction history (Zheng et al., 2023; Cao et al., 2024; Oh et al., 2024) or user reviews about items (Kim et al., 2024). These preferences are then used as targets for auxiliary tasks (Cao et al., 2024; Zheng et al., 2023), instructions for retrieval (Oh et al., 2024), or step-by-step reasoning (Kim et al., 2024). Incorporating such information usually improves recommendation performance.

---

[1]Code is available at `https://github.com/facebookresearch/preference_discerning`.
[2]†: Corresponding author.

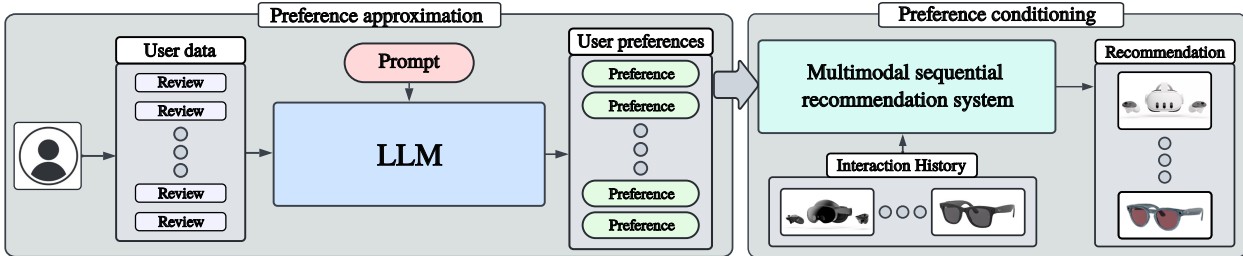

Figure 1: The preference discerning paradigm consists of two phases: *preference approximation* and *preference conditioning.* In preference approximation phase, a pre-trained LLM is used to infer user preferences from user-specific data. In preference conditioning phase, a sequential recommendation model is conditioned on the generated user preferences, enabling personalized recommendations.

Current sequential recommendation models lack the ability to dynamically adapt to changing user preferences after training, as they rely solely on the past interaction history of a user. Consider a scenario in which a user interacts on a social media platform and primarily views certain contents. Current recommendation models will continue to recommend similar content as no other information is provided to them. However, the user's interest may change over time influenced by lifestyle changes, career transitions, hobbies, or life events. For example, a user who uses social media for entertainment might start learning a new skill, but still receives viral videos instead of tutorials. This issue may be mitigated by users providing their preferences to the recommendation model; however, this ability is lacking in available models. To adapt to such situations, they require re-training after the user interacts with different items. Furthermore, there is a lack of understanding to what extent current recommendation models accurately discern user preferences.

To address these limitations, we propose a novel paradigm, which we term *preference discerning.* The aim of preference discerning is to approximate user preferences from previous comments, reviews, or recent activity and to provide them to the recommendation model, such that it can dynamically adapt its recommendations. Preference discerning consists of two stages: (1) preference approximation and (2) preference conditioning. In the first stage, we use LLMs to distill user- and item-specific data into short and concise user preferences. The second stage trains a sequential recommendation model conditioned on the generated preferences in its context to generate item recommendations. This in-context conditioning unlocks steering via generated user preferences, effectively combining the sequential prior from interaction history with the user preferences. This allows users to specify the item properties they wish to avoid or prefer in natural language. The model then integrates this information with previous interactions to dynamically adapt the recommendation.

To evaluate preference discerning capabilities of sequential recommendation models, we propose a holistic benchmark that comprises five evaluation axes: (1) preference-based recommendation, (2) sentiment following, (3) fine-grained steering, (4) coarse-grained steering, and (5) history consolidation. We evaluate state-of-the-art generative retrieval methods on our benchmark and find that they lack several key abilities of preference discerning. Therefore, we introduce a novel multimodal generative retrieval method named **M**ixedmodal prefer**en**ce **d**iscer**ner** (Mender) that effectively fuses pre-trained language encoders with the generative retrieval framework (Rajput et al., 2023) for preference discerning. We demonstrate that preference discerning capabilities can naturally emerge when training solely on preference-based recommendation data. Furthermore, we show that preference discerning capabilities can be obtained by augmenting the training data with training splits for the different axes. As a result, Mender can be effectively steered by different user preferences provided in its context to recommend specific items. Ultimately, Mender outperforms the existing state-of-the-art generative retrieval models on most evaluation axes of our benchmark. In summary, our contributions are as follows.

- We introduce a novel paradigm called *preference discerning*, where the generative recommendation model is conditioned on user preferences within its context.
- We propose a comprehensive benchmark for evaluating preference discerning, comprising of five distinct evaluation scenarios that provide a holistic assessment of its capabilities.

- We present Mender, a multimodal baseline that integrates collaborative semantics with language preferences, achieving state-of-the-art performance on our proposed benchmark.

## 2 Related Work

**Sequential Recommendation** can be categorized into two major scenarios: search (Nigam et al., 2019) and recommendation (Covington et al., 2016). The former assumes access to a query from a user that reflects their intent (He et al., 2022), whereas the latter scenario does not make such an assumption. For the recommendation scenario, numerous works have investigated the use of additional information to enhance recommendation performance (Meng et al., 2020; Hidasi et al., 2016; Liu et al., 2021; Zhang et al., 2019a; Bogina & Kuflik, 2017; Li et al., 2020). Our work introduces a new paradigm that enables in-context steering of sequential recommendation models through textual user preferences.

**Existing Benchmarks** for recommendation vary in their representation of user preferences and the tasks they evaluate. Oh et al. (2024) proposed a benchmark for instruction-following in information retrieval where instructions are generated from search queries. The C4 benchmark (Hou et al., 2024) uses complex search queries that reflect user preferences for retrieval. Contrary, we focus on user preferences in sequential recommendation. Such preferences are often modeled indirectly from user queries and responses to recommended items (Min et al., 2023; Huang et al., 2013; Ma et al., 2018), or represented as edges on graphs (Ying et al., 2018; Li et al., 2019). In query-aware sequential recommendation He et al. (2022) the model is given keywords in its context that represent the user's intent but do not capture their preferences. In contrast, our benchmark builds on established datasets (Ni et al., 2019; Kang & McAuley, 2018) and augments them with generated user preferences to evaluate preference discerning capabilities.

**Generative Retrieval** uses autoregressive modeling to generate the next item, rather than performing pairwise comparisons between a user representation and all item representations. The promise of generative retrieval is efficient operation on industrial-scale item sets (Singh et al., 2024). Therefore, in our work we focus mainly on applying preference discerning to generative retrieval. Rajput et al. (2023) proposes tokenizing items in the form of semantic IDs (Lee et al., 2022) instead of random IDs. The benefit of this approach is that very large item sets can be represented as a combination of ids that reflect their semantic similarity. Subsequent works have investigated the effect of learned tokenization (Sun et al., 2023) and additional objectives (Li et al., 2024; Wang et al., 2024). Our Mender represents items as semantic IDs and fuses them with pre-trained LMs to effectively steer the recommendation.

**Language-Based Sequential Recommendation** rely on the premise of enhanced transparency and actionable interrogation of recommendation systems (Radlinski et al., 2022). Furthermore, language provides a natural interface for users to express their preferences. Recent works have used LLMs to approximate user preferences by representing user and item specific data in natural language (Zheng et al., 2023; Oh et al., 2024; Cao et al., 2024), by conditioning the LLM on user embeddings (Ning et al., 2024), or by leveraging user reviews of items (Kim et al., 2024). In this context, Kang et al. (2023) found that an effective preference approximation may require fine-tuning of the LLM. Other studies have explored the use of LLMs for data augmentation in sequential recommendation (Geng et al., 2022; Zhang et al., 2019b; Luo et al., 2024). In the near-cold start scenario, Sanner et al. (2023) demonstrated that user preferences represented in natural language can be particularly effective. Li et al. (2023) showed the benefit of moving from ID-based representations to text-based representations of the interaction history. Similarly, Petrov & Macdonald (2023) represents all items in natural language and performs a ranking conditioned on past interactions. Zheng et al. (2023) explored aligning semantic IDs with natural language by adding auxiliary tasks.

## 3 Methodology

We first elaborate on the task of sequential recommendation and give important background in Section 3.1. In addition, we elaborate on the core components of preference discerning, namely *preference approximation* and *preference conditioning* (see Fig. 1). We cover our preference approximation pipeline in Section 3.2 and elaborate on our proposed preference-conditioned method Mender in Section 3.3 and Section 3.4. Finally, we elaborate on the construction of our benchmark to evaluate preference discerning capabilities in Section 3.5.

### 3.1 Background

In sequential recommendation, the goal is to provide personalized recommendation for users based on their interaction history. To this end, we assume access to a set of users $\mathcal{U}$ and a set of items $\mathcal{I}$. For each user $u \in \mathcal{U}$, we assume access to a sequence of item interactions in chronological order: $s_u = \left[ i_u^{(1)}, \ldots, i_u^{(T_u)} \right]$, where $T_u$ represents the time horizon of a particular user $u$ who has interacted with items $i_u \in \mathcal{I}$. The task of sequential recommendation is then to predict item $i_u^{(T_u)}$ given $\left[ i_u^{(1)}, \ldots, i_u^{(T_u-1)} \right]$. Traditional sequential recommendation systems (Kang & McAuley, 2018; Zhou et al., 2020; Sun et al., 2019; Hidasi et al., 2016) are based on sequence modeling architectures (Hochreiter & Schmidhuber, 1997; Devlin et al., 2019; Vaswani et al., 2017) to represent users and items via dense embeddings. These embeddings are then leveraged to retrieve the most relevant items for a user via pariwise comparisons using maximum inner product search. This approach can be computationally expensive as the number of items grows and each item must be represented as an embedding to be stored. Generative retrieval (Rajput et al., 2023) aims at alleviating the need for pairwise comparisons and storing unique embeddings for each item by leveraging semantic IDs (Lee et al., 2022) in combination with generative modeling. These approaches have proven to be effective on industrial-scale item sets (Singh et al., 2024).

### 3.2 Preference Approximation

Preference approximation refers to the process of distilling user- and item-specific data into short and concise user preferences. This compression is crucial, as sequential recommendation models are usually limited by the amount of information that can be provided in their input. This information may include user reviews, profiles, posts, demographic information, or any other relevant details. Furthermore, incorporating item-specific information is crucial, as it provides additional context that can help alleviate the vagueness or incompleteness often encountered in user-specific data. Preference approximation is a necessary prerequisite that enables in-context conditioning on the generated user preferences.

To approximate user preferences, we assume access to user-specific data including user reviews $r_u \in \mathcal{R}$ and descriptions of items in natural language. For each user $u$ and for each timestep $1 \le t \le T_u$, we collect reviews $\{r_u^{(1)}, \ldots, r_u^{(t)}\}$ along with item information $\{i_u^1, \ldots, i_u^{(t)}\}$ from their interaction history $s_u$ and prompt an LLM to approximate the user's preferences. We add a prompt $x$ (see Appendix C) to the interaction history that contains general instructions such as ignoring aspects such as

---

**Algorithm 1** Preference Approximation

**Input:** prompt $x$, users $\mathcal{U}$, items $\mathcal{I}$, reviews $\mathcal{R}$, Language Model LLM$(\cdot)$, user sequence length $T_u$

1: **for** $u \in \mathcal{U}$ **do**
2:      **for** $t \in \{1, \ldots, T_u\}$ **do**
3:         $\mathcal{P}_u^{(t)} \leftarrow \text{LLM}\left( \left[ x; i_u^{(1)}; r_u^{(1)}; \ldots; i_u^{(t)}; r_u^{(t)} \right] \right)$
4:      **end for**
5: **end for**

---

delivery time or pricing, and encode aversions of the user. With this process, we obtain a set of five user preferences $\mathcal{P}_u^{(t)}$ for each timestep $t$ based on past interactions. Importantly, the information contained in the different user preferences in $\mathcal{P}_u^{(t)}$ is mostly orthogonal, i.e., each preference refers to different items or item properties (see an example in Appendix C). To verify the quality of the generated preferences, we conduct a manual confirmation study (see Appendix F). The participants found that around 75% of the generated preferences correctly approximate the user's preferences. A schematic illustration of the preference generation procedure is shown in Fig. 10 along with pseudocode in Algorithm 1. For details on prompts, the generation process, or the granularity of preferences, we refer to Appendix C.

### 3.3 Multimodal Preference Discerner (Mender)

Dynamically adapting to evolving user preferences requires the recommendation model itself to be conditioned on them, which leads to recommendation steerability. Steerability in this context can be defined as guiding a recommendation model towards or away from certain items, based on the provided preferences and their given context. For example, if user preferences are given as "user prefers AR devices prioritizing

comfort and minimal eye strain" (see Figure 10), the recommendation model must steer its recommendations towards items relevant in this context. To achieve steerability, we perform preference conditioning.

In preference conditioning, we explicitly provide generated user preferences as an additional input to the sequential recommendation model. This requires the model to be capable of processing natural language input and to predict item identifiers. Therefore, we design a recommendation model that efficiently fuses the generated user preferences with item descriptions in its context and predicts item identifiers. This results in our new method, Mender, a novel multimodal generative sequential recommendation model. Mender builds on the TIGER (Rajput et al., 2023), a generative retrieval model trained using semantic IDs. These semantic IDs are obtained by training a RQ-VAE (Lee et al., 2022) on embeddings of items in the Sentence-T5 space. Given an item embedding $e \in \mathbb{R}^d$, the RQ-VAE quantizes $e$ into a discrete feature map as:

$$\mathcal{RQ}(e, \mathcal{C}, N) = (k_1, \ldots, k_N) \in [K]^N \tag{1}$$

where $\mathcal{C}$ represents a finite set of tuples $\{(k, c_k)\}_{k \in K}$, $K$ denotes the granularity of the codebook $\mathcal{C}$ with embeddings $\{c_k | 1 \leq k \leq K\}$, and $N$ corresponds to the depth of the RQ-VAE, i.e., the number of codebooks. A user sequence $s_u$ is then represented as a sequence of semantic IDs: $\left[k_1^{(1)}, \ldots, k_N^{(1)}, \ldots, k_1^{(T_u)}, \ldots, k_N^{(T_u)}\right]$, which serves as input to train a Transformer model (Vaswani et al., 2017). To enable conditioning on natural language, we leverage pre-trained language encoders. Specifically, we represent both the interaction history and the user preference in natural language and process them with a pre-trained FLAN-T5-Small encoder (Chung et al., 2024). This is inspired by Li et al. (2023); Paischer et al. (2022; 2023), who demonstrated the benefits of history compression using natural language. The decoder of Mender is randomly initialized and conditioned on the language encoder through cross-attention to predict semantic IDs. Since semantic IDs are represented in a separate embedding space than natural langauge, Mender can be classified as multimodal.

## 3.4 Mender Variants

To strike a balance between efficiency and performance, we further propose two variants of Mender, namely Mender$_{\text{Tok}}$ and Mender$_{\text{Emb}}$. The key difference between these variants lies in the way they encode user preferences and items. Mender$_{\text{Tok}}$ encodes user preferences and items as a single sequence of language tokens. Hence, this model provides a performant and high-capacity model with strong language understanding capabilities. However, Mender$_{\text{Tok}}$ processes the entire token sequence at once, making it suitable for complex language-based in-context learning and fine-tuning. In contrast, Mender$_{\text{Emb}}$ encodes each user preference and item separately using a pre-trained embedding model from Su et al. (2023). Mender$_{\text{Emb}}$ allows pre-computing item and preference embeddings, resulting in improved training efficacy. Mender$_{\text{Emb}}$ does not require fine-tuning, as propagating through the embedding model for each preference/item is prohibitively expensive.

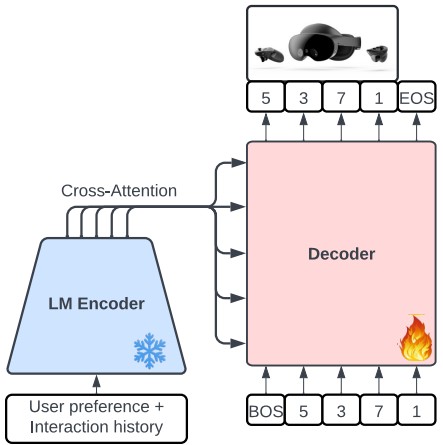

Figure 2: Mender. The decoder generates semantic IDs conditioned on user preferences and interactions via cross-attention with a pre-trained language encoder.

## 3.5 Evaluating Steerability via User Preferences

Steerability can come in various forms, and there is no conventional way to evaluate it. In this work, we define five performance axes that shall evaluate different aspects of steerability (see Fig. 3):

- **Preference-based Recommendation**: recommending the correct item based on user preference and interaction history
- **Fine & Coarse-Grained Steering**: The ability to steer towards items that are similar (fine-grained) or very distinct (coarse-grained) to items observed during training
- **Sentiment Following Capabilities**: the ability to understand the sentiment in steering (e.g, user likes something vs. user does NOT like something)
- **History Consolidation**: the ability of the model to incorporate multiple user preferences.

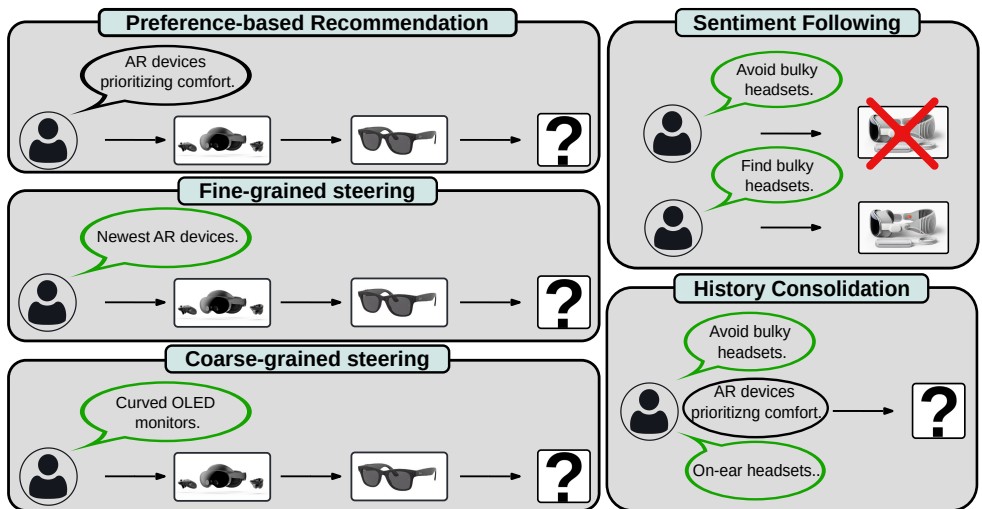

Figure 3: Five evaluation axes for preference discerning we focus on in this work: Preference-based Recommendation, Fine-grained steering, Coarse-grained steering, Sentiment following, and History Consolidation. Preferences highlighted in green indicates that they are unseen during training.

We provide a more in-depth definition of these performance axes in the following paragraphs.

**Preference-based Recommendation.** This evaluation scenario extends the sequential recommendation scenario by incorporating the generated user preferences. For this task, the model receives a single user preference of the set $\mathcal{P}_u^{(t-1)}$ along with the interaction history and must predict the next item $i_u^{(t)}$. We select the preference that yields the maximum cosine similarity to $i^{(t)}$ in a pre-trained sentence embedding space (Ni et al., 2022). More formally, given a pre-trained sentence embedding model $\phi(\cdot)$, we select $p_u^{(t)}$ as

$$p_u^{(t)} = \arg \max_{p \in \mathcal{P}_u^{(t-1)}} \frac{\phi(p)^\top \phi(i_t)}{\|\phi(p)\|\|\phi(i_u^{(t)})\|}. \tag{2}$$

This results in a setting where each ground truth item $i_u^{(t)}$ is associated with a single user preference $p_u^{(t)}$. Therefore, the input to the sequential recommendation system is a sequence of $\left[p_u^{(t)}, i_u^{(1)}, \dots, i_u^{(t-1)}\right]$ and the task is to predict $i_u^{(t)}$. Since $p_u^{(t)}$ is generated based only on information about past items ($p \in \mathcal{P}_u^{(t-1)}$), there is no information leak that could reveal the ground truth item, that is, there is no information leak and the underlying aleatoric uncertainty of the task is preserved.

**Fine-Grained & Coarse-Grained Steering.** To evaluate the ability of the model to discern user preferences that are semantically related to items, we introduce the tasks of fine- and coarse-grained steering. Recall that the preference-based recommendation scenario captures the underlying uncertainty of the original recommendation task as we provide the model with $p_u^{(t-1)}$ to predict $i_u^{(t)}$. This can result in cases where $p_u^{(t-1)}$ is not semantically related to $i_u^{(t)}$, since often $i_u^{(t)}$ is not related to previously acquired items. Therefore, our goal is to quantify the model's ability to be steered towards items that are either semantically similar or very distinct from $i_u^{(t)}$ by modifying the user preference in its context. To achieve this, we identify a very similar item $\tilde{i}^{(t)}$ and a very distinct item $\hat{i}_u^{(t)}$ to the ground-truth item $i_u^{(t)}$ by

$$\tilde{i}_u^{(t)} = \arg \max_{i \in \mathcal{I} \setminus \{i_u^{(t)}\}} \frac{\phi(i)^\top \phi(i_u^{(t)})}{\|\phi(i)\|\|\phi(i_u^{(t)})\|}, \quad \text{and} \quad \hat{i}_u^{(t)} = \arg \min_{i \in \mathcal{I} \setminus \{i_u^{(t)}\}} \frac{\phi(i)^\top \phi(i_u^{(t)})}{\|\phi(i)\|\|\phi(i_u^{(t)})\|}. \tag{3}$$

Next, we associate $\tilde{i}^{(t)}$ and $\hat{i}_u^{(t)}$ with different user preferences $p_1$ and $p_2$ by

$$p_1 = \arg \max_{p \in \mathcal{P}} \frac{\phi(p)^\top \phi(\tilde{i}^{(t)})}{\|\phi(p)\|\|\phi(\tilde{i}^{(t)})\|}, \quad \text{and} \quad p_2 = \arg \max_{p \in \mathcal{P}} \frac{\phi(p)^\top \phi(\hat{i}_u^{(t)})}{\|\phi(p)\|\|\phi(\hat{i}_u^{(t)})\|}, \tag{4}$$

where $\mathcal{P}$ denotes the set of accumulated preferences across all users and items. Additionally, we obtain a target user $\hat{u}$ with the same ground truth item $i_{\hat{u}}^{(t)} = i_u^{(t)}$, but a different interaction history. The motivation for this is to enhance the variability in the generated datasets. By combining these elements, we create two new sequences: $\left[p_1, i_{\hat{u}}^{(1)}, \ldots, i_{\hat{u}}^{(t-1)}\right]$ and $\left[p_2, i_u^{(1)}, \ldots, i_u^{(t-1)}\right]$ with ground-truth items $\tilde{i}_u^{(t)}$ and $\hat{i}_u^{(t)}$, respectively. A visual illustration of this procedure is provided in Fig. 13. Throughout the dataset creation process, we ensure that the preferences used during training are not associated with the evaluation items. This allows us to evaluate the model's ability to generalize to new preferences not observed during training.

**Sentiment Following.** An important aspect of preference discerning is whether the recommendation model comprehends the sentiment of user preferences. A user may provide negative preferences about items or properties that should be avoided To assess the ability of recommendation models to identify and follow the sentiment of issued preferences, we attempt to establish a causal relationship between items that received a negative review and negative preferences generated during the preference approximation stage. The intuition is that a negative user preference generated by the LLM is likely caused by an item in $s_u$ that received a negative review. Therefore, we first need to identify negative reviews and user preferences. This is done using a pre-trained sentiment classifier (Hartmann et al., 2023). Then we match each negative user preference $p_u^-$ at timestep $t$ with a negative review in $\{r_u^{(j)} | 1 \leq j \leq t\}$. The matching is again done via cosine similarity in a pre-trained Sentence-T5 space (Ni et al., 2022) This results in tuples $(p_u^-, i)$, where $p_u^-$ represents a negative preference and $i$ denotes the item that received the negative review to which the preference was matched. To obtain a positive pair $(p_u^+, i)$, we apply a rule-based inversion heuristic to the negative preference (see Appendix D for details). The compiled data consist solely of $(p_u^{\pm}, i)$ tuples without interaction history. During evaluation we provide the preference in the context of the model and it should either predict or *not* predict the item based on whether the provided preference is positive or negative.

To evaluate sentiment following, we rely on a combined hit-rate measure. Given a set of k predicted candidate items $\mathcal{C} = \{\bar{i}_1, \ldots, \bar{i}_k\}$, we check whether the ground truth item occurs in $\mathcal{C}$ (that is, $\mathbb{1}_{\mathcal{C}}(i) = 1$, where $\mathbb{1}(\cdot)$ represents the indicator function). In practice, we obtain two sets of predictions $\mathcal{C}^+$ and $\mathcal{C}^-$, where $\mathcal{C}^+$ is obtained using positive preference $p_u^+$ and $\mathcal{C}^-$ using the negative preference $p_u^-$ for item $i$. The combined hit rate measure is computed as $m = \mathbb{1}_{\mathcal{C}^+}(i) \wedge \neg \mathbb{1}_{\mathcal{C}^-}(i)$ where $m = 1$ indicates that the model successfully recovered the item for $p_u^+$, while simultaneously *not* predicted it for $p_u^-$. This measure can be calculated for different sizes $(k)$ of prediction sets as $m@k$.

**History Consolidation.** A user may have multiple preferences about different items. In such a case, the sequential recommendation system must incorporate multiple user preferences to steer their recommendation. Therefore, our aim is to evaluate the ability of the system to incorporate multiple user preferences, some of which are potentially not related to the next item. To simulate this, we simultaneously provide the model with the five generated preferences $\mathcal{P}_u^{(t-1)}$ to predict the ground-truth item $i_u^{(t)}$. This evaluation scenario can be considered more difficult than preference-based recommendation, as it potentially has a higher noise ratio in the provided preferences. In this evaluation scenario, the preference originally matched is contained in the set of accumulated user preferences $\mathcal{P}$. Therefore, in order to accurately predict the ground truth item, the model must infer the matched preference from $\mathcal{P}$. The corresponding evaluation sequences are structured as $\left[p_{u_1}^{(T_u-1)}, \ldots, p_{u_5}^{(T_u-1)}, i_1, \ldots, i_u^{(T_u-1)}\right]$ and contain all five generated user preferences.

# 4 Experiments

We evaluate our approach on four widely used datasets, namely three Amazon reviews subsets (Ni et al., 2019) and Steam (Kang & McAuley, 2018). An overview of the dataset statistics can be found in Table 3 in Appendix B. To generate user preferences, we utilize the `LlaMa-3-70B-Instruct`[2] model. For the sentiment classification, we employ the model trained by Hartmann et al. (2023)[3]. The resulting preference statistics, including the number of generated preferences, the proportion of positive and negative preferences, and the sample sizes for each evaluation split, are presented in Table 4. Our data generation pipeline is built entirely on open-source models, making it easily extensible to additional datasets.

---

[2]https://huggingface.co/meta-llama/Meta-Llama-3-70B-Instruct
[3]https://huggingface.co/siebert/sentiment-roberta-large-english

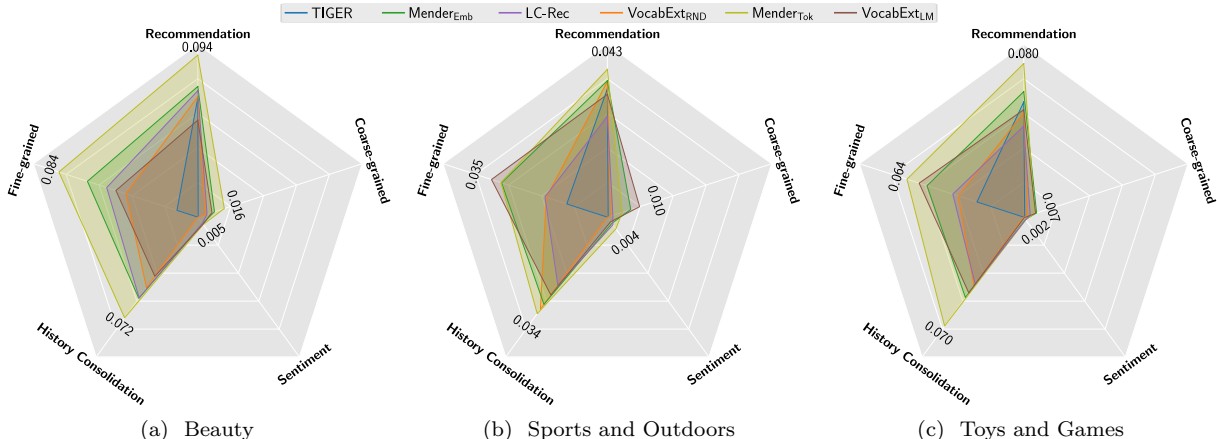

Figure 4: Recall@10 for all methods on our novel benchmark, evaluating preference discerning across three subsets of the Amazon review dataset: Beauty (4a), Sports and Outdoors (4b), and Toys and Games (4c). Mender$_{Tok}$ mostly outperforms generative retrieval competitors across *Recommendation*, *Fine-grained steering* and *History consolidation*. All methods struggle on *Sentiment following* and *Coarse-grained steering*.

For training our models, we use the preference-based recommendation data, which consists of a single user preference and the interaction history. Unless mentioned otherwise, the additional generated data splits (positive/negative and fine/coarse data) are used solely for evaluation purposes. Following (Rajput et al., 2023), we limit the maximum number of items in a user sequence to the 20 most recent ones. For the Beauty, Toys and Games, and Steam datasets, we found it beneficial to also fine-tune the language encoder, for which we use LoRA (Hu et al., 2022). By default, we use the FLAN-T5-Small (Chung et al., 2024) language encoder for Mender$_{Tok}$. We adopt a leave-last-out data split, where the penultimate item of a sequence is used for validation and the last item is used for testing (Kang & McAuley, 2018; Sun et al., 2019). Our evaluation benchmark is based only on the validation and test items of that split along with their paired user preferences, that is, we use preferences for training and inference. The remaining items in each sequence are used for training, except for the first item, since no user preferences are available for it. We evaluate our trained baselines using common retrieval metrics, including Recall (or Hit Rate), and Normalized Discounted Cumulative Gain (Järvelin & Kekäläinen, 2002, NDCG). Implementation details for training the RQ-VAE and Transformer models can be found in Appendix A.1 and Appendix A.2, respectively. All our methods are trained on single A100 or V100 GPUs using PyTorch (Paszke et al., 2019).

## 4.1 Baselines

We train and evaluate a range of generative retrieval baselines and compare their performance to Mender.

**TIGER** (Rajput et al., 2023) is a state-of-the-art generative retrieval model based on semantic IDs. Although TIGER is not conditioned on user preferences, we still evaluate its performance on our benchmarks for recommendation, fine-grained steering, and coarse-grained steering. The latter two essentially evaluate how well TIGER predicts a very similar or distinct item to the ground-truth item.

**VocabExt$_{RND}$** is based on extending the vocabulary of the TIGER model, which allows it to be conditioned on language preferences. Notably, this version does not leverage any pre-trained components.

**LC-Rec** (Zheng et al., 2023) extends the vocabulary of a pre-trained LM with newly initialized embeddings that represent semantic IDs. We fine-tune the LM using LoRA (Hu et al., 2022), but do not add the auxiliary tasks. Additionally, we reduce the dimensionality of the language model head to match the number of semantic IDs, as language generation is not required for our task.

**VocabExt$_{LM}$** represents the past interaction history in language as done for Mender$_{Tok}$ and Mender$_{Emb}$, but initializes the decoder with a pre-trained language decoder. Therefore, this baseline operates on the same semantic gap as the Mender variants. We again leverage LoRA for fine-tuning.

## 4.2 Results

We present a detailed analysis of the results obtained by the different methods on our benchmark for three subsets of Amazon reviews (Beauty, Sports and Outdoors, and Toys and Games) and Steam datasets. Fig. 4 and Fig. 5a show Recall@10 for all methods on the Amazon and Steam datasets, respectively. Table 1 also shows Recall@10 plus additional metrics, such as Recall@5, NDCG@5, NDCG@10, as well as relative improvements of Mender over the best baseline method. In Appendix E, we report the corresponding standard deviations for all methods on all datasets. Our results reveal several key trends: (i) incorporating preferences consistently improves performance; (ii) training on preference-based recommendation data leads to the emergence of fine-grained steering on certain datasets; (iii) current models struggle with sentiment following; and (iv) both coarse-grained steering and sentiment following can be achieved through data augmentations. Additionally, we provide ablation studies on data mixtures and the impact of adding user preferences.

Table 1: Performance for all methods on all evaluation axes for all datasets trained on recommendation data. We report average performance across three random seeds as well as relative improvements of Mender to the second-best performing baseline and highlight best performance in boldface. For sentiment following we report $m@k$ for $k \in \{5, 10\}$ instead of Recall@k.

| Methods | Sports and Outdoors | | | | Beauty | | | | Toys and Games | | | | Steam | | | |
|---|---|---|---|---|---|---|---|---|---|---|---|---|---|---|---|---|
| | Recall @5 | NDCG @5 | Recall @10 | NDCG @10 | Recall @5 | NDCG @5 | Recall @10 | NDCG @10 | Recall @5 | NDCG @5 | Recall @10 | NDCG @10 | Recall @5 | NDCG @5 | Recall @10 | NDCG @10 |
| Recommendation | | | | | | | | | | | | | | | | |
| TIGER | 0.0249 | 0.0158 | 0.0377 | 0.0199 | 0.0431 | 0.0275 | 0.0681 | 0.0356 | 0.0375 | 0.0238 | 0.0600 | 0.0311 | 0.1630 | **0.1440** | 0.1930 | 0.1530 |
| VocabExt$_{RND}$ | 0.0238 | 0.0151 | 0.0392 | 0.0201 | 0.0434 | 0.0277 | 0.0697 | 0.0362 | 0.0330 | 0.0205 | 0.0544 | 0.0275 | 0.1660 | 0.1420 | 0.2000 | 0.1540 |
| LC-Rec | 0.0195 | 0.0124 | 0.0291 | 0.0156 | 0.0457 | 0.0294 | 0.0731 | 0.0382 | 0.0327 | 0.0209 | 0.0473 | 0.0256 | 0.1600 | 0.1370 | 0.1940 | 0.1480 |
| VocabExt$_{LM}$ | 0.0233 | 0.0148 | 0.0355 | 0.0187 | 0.0345 | 0.0224 | 0.0561 | 0.0293 | 0.0371 | 0.0234 | 0.0559 | 0.0296 | 0.1547 | 0.1305 | 0.1878 | 0.1412 |
| Mender$_{Emb}$ | 0.0264 | 0.0173 | 0.0394 | 0.0215 | 0.0494 | 0.0321 | 0.0755 | 0.0405 | 0.0422 | 0.0267 | 0.0653 | 0.0342 | 0.1450 | 0.1110 | 0.1820 | 0.1230 |
| Mender$_{Tok}$ | **0.0282** | **0.0188** | **0.0427** | **0.0234** | **0.0605** | **0.0401** | **0.0937** | **0.0508** | **0.0533** | **0.0346** | **0.0799** | **0.0432** | **0.1680** | **0.1440** | **0.2040** | **0.1560** |
| Rel. Impr. | +13.2% | +18.9% | +8.9% | +16.4% | +32.4% | +36.4% | +28.1% | +33.0% | +42.1% | +45.4% | +33.2% | +38.9% | +1.2% | +0.0% | +2.0% | +1.3% |
| Fine-grained steering | | | | | | | | | | | | | | | | |
| TIGER | 0.0061 | 0.0037 | 0.0118 | 0.0055 | 0.0119 | 0.0074 | 0.0195 | 0.0098 | 0.0149 | 0.0092 | 0.0237 | 0.0120 | 0.0084 | 0.0052 | 0.0145 | 0.0072 |
| VocabExt$_{RND}$ | 0.0104 | 0.0063 | 0.0186 | 0.0089 | 0.0229 | 0.0163 | 0.0437 | 0.0220 | 0.0200 | 0.0123 | 0.0358 | 0.0174 | 0.0102 | 0.0064 | 0.0178 | 0.0088 |
| LC-Rec | 0.0119 | 0.0074 | 0.0190 | 0.0097 | 0.0348 | 0.0218 | 0.0563 | 0.0288 | 0.0248 | 0.0153 | 0.0388 | 0.0198 | 0.0157 | 0.0098 | 0.0264 | 0.0133 |
| VocabExt$_{LM}$ | **0.0214** | **0.0132** | **0.0352** | **0.0176** | 0.0292 | 0.0186 | 0.0498 | 0.0253 | 0.0341 | 0.0220 | 0.0572 | 0.0294 | **0.0217** | 0.0133 | **0.0365** | **0.0180** |
| Mender$_{Emb}$ | 0.0173 | 0.0106 | 0.0322 | 0.0154 | 0.0276 | 0.0174 | 0.0465 | 0.0234 | 0.0316 | 0.0199 | 0.0529 | 0.0267 | 0.0184 | 0.0114 | 0.0287 | 0.0147 |
| Mender$_{Tok}$ | 0.0190 | 0.0117 | 0.0324 | 0.0159 | **0.0534** | **0.0344** | **0.0844** | **0.0444** | **0.0378** | **0.0237** | **0.0639** | **0.0321** | 0.0211 | **0.0134** | 0.0352 | 0.0179 |
| Rel. Impr. | -12.6% | -12.8% | -8.6% | -10.7% | +53.4% | +57.8% | +49.9% | +54.2% | +10.9% | +7.7% | +11.7% | +9.2% | -2.8% | +1% | -3.7% | -1% |
| Coarse-grained steering | | | | | | | | | | | | | | | | |
| TIGER | 0.0001 | 0.0000 | 0.0003 | 0.0001 | 0.0003 | 0.0001 | 0.0003 | 0.0002 | 0.0003 | 0.0001 | 0.0006 | 0.0003 | 0.0005 | 0.0003 | 0.0008 | 0.0004 |
| VocabExt$_{RND}$ | 0.0005 | 0.0003 | 0.0010 | 0.0004 | 0.0023 | 0.0014 | 0.0046 | 0.0021 | 0.0013 | 0.0009 | 0.0021 | 0.0011 | 0.0032 | 0.0018 | 0.0055 | 0.0026 |
| LC-Rec | 0.0010 | 0.0006 | 0.0017 | 0.0009 | 0.0032 | 0.0019 | 0.0059 | 0.0028 | 0.0022 | 0.0013 | 0.0036 | 0.0017 | 0.0028 | 0.0018 | 0.0049 | 0.0024 |
| VocabExt$_{LM}$ | **0.0047** | **0.0028** | **0.0098** | **0.0044** | 0.0053 | 0.0033 | 0.0086 | 0.0044 | **0.0037** | **0.0022** | 0.0065 | 0.0030 | **0.0047** | **0.0029** | 0.0077 | 0.0039 |
| Mender$_{Emb}$ | 0.0036 | 0.0022 | 0.0071 | 0.0033 | 0.0057 | 0.0035 | 0.0101 | 0.0050 | 0.0035 | 0.0021 | **0.0071** | **0.0032** | 0.0042 | 0.0024 | 0.0067 | 0.0032 |
| Mender$_{Tok}$ | 0.0023 | 0.0013 | 0.0045 | 0.0021 | **0.0094** | **0.0059** | **0.0161** | **0.0080** | 0.0032 | 0.0020 | 0.0060 | 0.0029 | 0.0043 | 0.0027 | **0.0081** | **0.0040** |
| Rel. Impr. | -30.6% | -27.3% | -38.1% | -33.3% | +77.4% | +78.8% | +87.2% | +81.8% | -15.6% | -4.8% | +9.2% | +6.7% | -9.3% | -7.4% | +5.2% | +2.6% |
| Sentiment following | | | | | | | | | | | | | | | | |
| TIGER | 0.0000 | - | 0.0000 | - | 0.0000 | - | 0.0000 | - | 0.0000 | - | 0.0000 | - | 0.0000 | - | 0.0000 | - |
| VocabExt$_{RND}$ | 0.0000 | - | 0.0000 | - | 0.0000 | - | 0.0000 | - | 0.0000 | - | 0.0000 | - | 0.0061 | - | 0.0086 | - |
| LC-Rec | 0.0018 | - | 0.0027 | - | 0.0029 | - | 0.0045 | - | 0.0008 | - | **0.0017** | - | 0.0033 | - | 0.0053 | - |
| VocabExt$_{LM}$ | 0.0019 | - | 0.0016 | - | 0.0027 | - | 0.0051 | - | 0.0012 | - | 0.0004 | - | 0.0049 | - | 0.0107 | - |
| Mender$_{Emb}$ | 0.0022 | - | 0.0022 | - | 0.0030 | - | 0.0047 | - | **0.0017** | - | 0.0015 | - | **0.0114** | - | **0.0185** | - |
| Mender$_{Tok}$ | **0.0035** | - | **0.0042** | - | **0.0043** | - | **0.0053** | - | 0.0016 | - | **0.0017** | - | 0.0084 | - | 0.0110 | - |
| Rel. Impr. | +84.2% | - | +55.6% | - | +48.3% | - | +3.9% | - | +41.7% | - | +0% | - | +86.9% | - | +72.9% | - |
| History consolidation | | | | | | | | | | | | | | | | |
| TIGER | 0.0000 | 0.0000 | 0.0000 | 0.0000 | 0.0000 | 0.0000 | 0.0000 | 0.0000 | 0.0000 | 0.0000 | 0.0000 | 0.0000 | 0.0000 | 0.0000 | 0.0000 | 0.0000 |
| VocabExt$_{RND}$ | 0.0190 | 0.0120 | 0.0329 | 0.0164 | 0.0303 | 0.0191 | 0.0504 | 0.0256 | 0.0260 | 0.0158 | 0.0441 | 0.0216 | 0.1366 | 0.1155 | 0.1642 | 0.1244 |
| LC-Rec | 0.0158 | 0.0101 | 0.0243 | 0.0129 | 0.0354 | 0.0226 | 0.0577 | 0.0297 | 0.0295 | 0.0185 | 0.0430 | 0.0229 | **0.1460** | **0.1277** | **0.1726** | **0.1363** |
| VocabExt$_{LM}$ | 0.0179 | 0.0112 | 0.0278 | 0.0145 | 0.0247 | 0.0155 | 0.0423 | 0.0211 | 0.0316 | 0.0195 | 0.0487 | 0.0251 | 0.0615 | 0.0440 | 0.0866 | 0.0521 |
| Mender$_{Emb}$ | 0.0206 | 0.0133 | 0.0312 | 0.0167 | 0.0352 | 0.0228 | 0.0580 | 0.0301 | 0.0314 | 0.0201 | 0.0516 | 0.0266 | 0.1241 | 0.0938 | 0.1558 | 0.1040 |
| Mender$_{Tok}$ | **0.0234** | **0.0151** | **0.0345** | **0.0187** | **0.0457** | **0.0304** | **0.0720** | **0.0388** | **0.0467** | **0.0302** | **0.0700** | **0.0377** | 0.0490 | 0.0317 | 0.0745 | 0.0399 |
| Rel. Impr. | +23.2% | +25.8% | +4.9% | +14.0% | +29.1% | +34.5% | +24.8% | +30.6% | +58.3% | +54.9% | +43.7% | +50.2% | -15.1% | -26.5% | -9.7% | -23.7% |

**Recommendation.** Our Mender$_{Tok}$ achieves the best performance on all datasets on the recommendation axis with relative improvements of up to 45%. The significant gap between TIGER and Mender$_{Tok}$ demon-

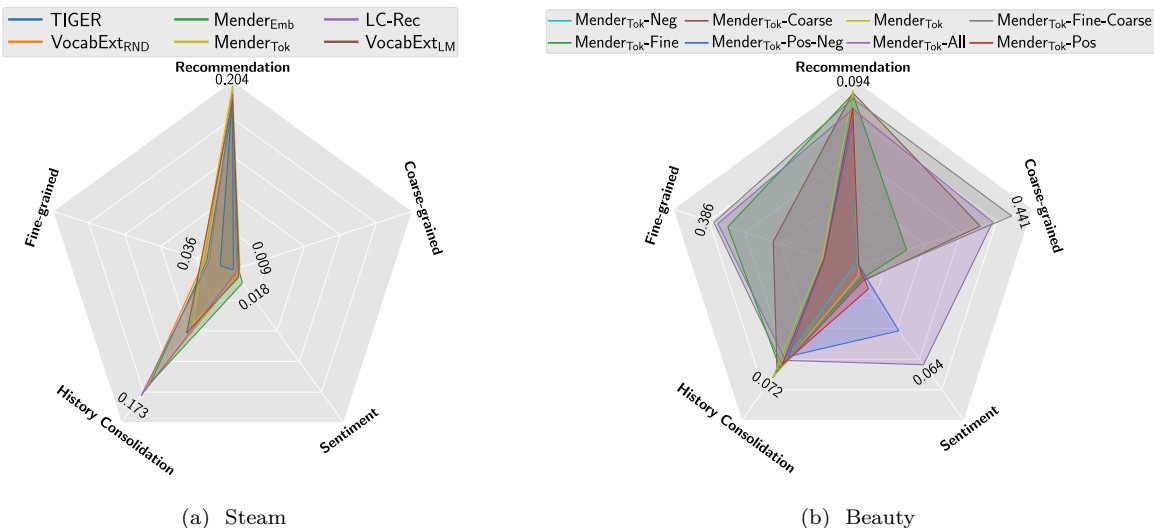

(a) Steam          (b) Beauty

Figure 5: Recall@10 of different baselines trained on the default recommendation data of the Steam dataset (5a) Mender$_{\text{Tok}}$ attains the highest performance on *Recommendation*, but all methods struggle on *Steering* and *Sentiment following*. 5b: Recall@10 for Mender$_{\text{Tok}}$ trained on different datasplits on the Amazon Beauty subset. Mender$_{\text{Tok}}$-All leverages training data augmentation resulting in a universal model that performs well across all axes of preference discerning.

strates the benefits of conditioning on the generated user preferences. Furthermore, we provide a comparison to traditional sequential recommendation baselines on the recommendation task in Table 2 in Appendix A.3 for the three Amazon subsets, which shows that our TIGER implementation outperforms those as well, and Mender further improves significantly on TIGER. In addition, Mender$_{\text{Emb}}$ performs second best in Amazon datasets, providing a decent trade-off between performance and training speed, reducing training time around five fold. Notably, other baselines such as VocabExt$_{\text{RND}}$ and LC-Rec sometimes perform worse than TIGER on Toys and Steam, indicating that they cannot properly align the semantic ID and language spaces. LC-Rec usually requires auxiliary tasks to align the two spaces properly (Zheng et al., 2023), while Mender successfully fuses them without training on auxiliary tasks. VocabExt$_{\text{RND}}$ performs significantly worse than both Mender versions due to its lack of a pre-trained language encoder, which requires learning the interaction between item history and user preferences from scratch. Based on these findings, we conclude that: (i) user preferences substantially enhance recommendation performance and (ii) representing both interaction history and preferences in natural language leads to performance improvements.

**Fine- and coarse-grained steering.** We observe that Mender$_{\text{Tok}}$ consistently achieves the best performance across all datasets for fine-grained steering with relative improvements of up to 70.5% to baselines. Interestingly, as illustrated in Fig. 4, fine-grained steering naturally emerges as a byproduct of training on preference-based recommendation data. However, this is not the case for the Steam dataset (Fig. 5a), where we notice a significant gap between the recommendation and the fine-grained steering performance. We surmise that the reason for this is the fundamental difference in the respective data distribution of the Amazon and Steam datasets. Prior work demonstrated that data distribution is an essential driving factor to elicit emerging capabilites such as in-context learning (Chan et al., 2022). We aim to verify this conjecture in future work. Finally, our results indicate that all methods struggle to perform coarse-grained steering, suggesting that preference-based recommendation data lacks a beneficial signal to facilitate its emergence.

**History Consolidation.** Generally, we observe that all methods attain lower scores on history consolidation compared to the recommendation. This is because the additional preferences are not necessarily related to the ground-truth item and thus add a substantial amount of noise. Furthermore, one of the five user preferences provided to the model contains information to identify the ground-truth item as they were matched during the benchmark generation. Therefore, the performance attained is a proxy for how well the model can identify a useful preference out of a set of potentially unrelated preferences. On the Amazon subsets, Mender$_{\text{Tok}}$

consistently attains the highest performance, while LC-Rec attains the best results on Steam. These findings suggest that preference-based methods can effectively fuse interaction history with multiple user preferences for recommendation. In Table 8 (Appendix E), we also show results for training on history consolidation data, demonstrating that this drastically degrades performance.

**Sentiment Following.** Although both Mender variants achieve the highest performance on different datasets, the overall performance on sentiment following is generally around an order of magnitude lower. This result indicates that all current models struggle with sentiment following. This finding presents an interesting avenue for future research that should prioritize the development of models that can accurately identify the sentiment of user preferences and adapt their retrieval accordingly. Prior works found that there is little to no gain in incorporating negative user preferences into recommendation models (Sanner et al., 2023). Our results confirm that current systems mostly lack the ability to discern negative preferences and to act accordingly. However, in the next section we show that this depends on *how* the negative preferences are used during training, and that it is indeed possible to obtain a model that improves along this axis.

### 4.3 Ablation Studies

**Importance of Preferences.** We perform an ablation study to investigate the impact of the generated user preferences. In Fig. 8 (Appendix A.4), we provide evidence that representing items in language instead of semantic IDs leads to improved rankings. Further, we train $\text{Mender}_{\text{Tok}}$ and (i) condition it only on preferences; (ii) condition it only on item descriptions; and (iii) condition it on both. We present our results for the Beauty dataset in Fig. 6. Our results clearly demonstrate the benefits of leveraging textual user preferences.

**Ablating Training Data Mixture to Study Steering emergence** Training on preference-based recommendation data does not elicit steering capabilities on Steam, or sentiment following capabilities on any dataset. Therefore, we aim to answer the question whether these capabilities can be elicited by directly training on additional data for steering and sentiment following. As can be seen in Table 3 in Appendix C, we also generate training splits during benchmark generation. Hence, we can answer this question by augmenting the preference-

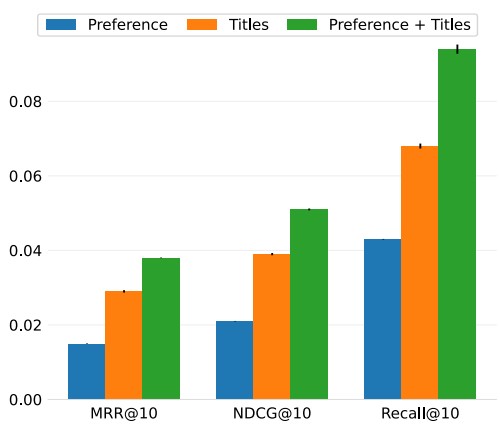

Figure 6: Ablation study highlighting the improvement obtained via combining items and user preferences in natural language.

based recommendation training set with the additional data sources and train different variants of $\text{Mender}_{\text{Tok}}$. Specifically, we train different versions of $\text{Mender}_{\text{Tok}}$ to study how steering-specific data impacts performance:

- **$\text{Mender}_{\text{Tok}}$-Pos/Neg**: uses only positive/negative preference-item pairs
- **$\text{Mender}_{\text{Tok}}$-Pos-Neg**: combines both positive and negative preference-item pairs
- **$\text{Mender}_{\text{Tok}}$-Fine/Coarse**: uses only fine/coarse-grained steering data
- **$\text{Mender}_{\text{Tok}}$-Fine-Coarse**: uses fine- and coarse-grained steering data
- **$\text{Mender}_{\text{Tok}}$-All**: trained on all data above.

When including the negative $(p_u^-, i)$ tuples, we simply minimize the likelihood and downweight it by a hyperparameter, as otherwise this term dominates the loss function. We present Recall@10 for Beauty in Fig. 5, right, and for Steam in Appendix E. We also report Recall@5, NDCG@5, and NDCG@10 for all methods and evaluation axes in Table 5 (Appendix E). Most importantly, coarse-grained steering and sentiment-following capabilities arise when we explicitly train the model on the respective data. Interestingly, $\text{Mender}_{\text{Tok}}$-All significantly improves on $\text{Mender}_{\text{Tok}}$ on all axes while maintaining performance on the recommendation axis. However, training on a data split in isolation improves over training on all data, i.e. $\text{Mender}_{\text{Tok}}$-Coarse leads to better coarse-grained steering than $\text{Mender}_{\text{Tok}}$-All, but lacks sentiment following. Furthermore, sentiment-following capabilities arise only when training jointly on positive *and* negative data. These findings present a fruitful avenue for future research on combining the different data sources.

**Scaling the language encoder.** By default, we use the FLAN-T5-Small encoder in our experiments. To investigate the effect of scaling the encoder, we compare the performance of $\text{Mender}_{\text{Tok}}$ with $\text{Mender}_{\text{Tok}}$-XXL, which uses the FLAN-T5-XXL encoder. The results of this experiment can be found in Table 9 (Appendix E). We find that the XXL variant drastically improves recommendation performance on Sports and Outdoors, fine- and coarse-grained steering on Toys and Games and Steam datasets, and sentiment following on all datasets. Furthermore, $\text{Mender}_{\text{Tok}}$-XXL drastically improves the history consolidation axis, which indicates that the observed gap in Fig. 5a can be attributed to the language encoder. This provides compelling evidence that more capable language encoders can lead to drastic improvements on the different performance axes.

## 5 Limitations

**Generalization.** The generalization capabilities of Mender are limited by the underlying generalization abilities of the language encoder. In Table 9 we presented results for larger variants of the FLAN-T5 encoder which show that there are gains to using more expressive LLMs, especially for axes such as coarse-grained steering or sentiment following. Moreover, recent work (Yang et al., 2025) showed that generalization to cold-start items is limited for generative retrieval methods. Mender might alleviate this problem by combining natural language with semantic IDs. Future work should investigate whether this is indeed the case.

**Computational complexity.** The computational complexity of Mender is mostly restricted by the model architecture. Currently, it suffers from the quadratic complexity of the Transformer (Vaswani et al., 2017) and the size of the language encoder that is used. $\text{Mender}_{\text{Emb}}$ partly alleviates this issue by pre-embedding items and user preferences, which usually results in slightly worse performance. However, there are fruitful alternatives that scale linearly with the sequence length (Dao & Gu, 2024) and provide a constant inference cost (Beck et al., 2024). We aim to investigate such architectures for Mender in the future.

**Preference Approximation.** Our preference approximation pipeline is computationally expensive, as it leverages LLMs with 70B parameters. We generated around 5M user preferences for the five different datasets, which requires massive parallelization of this pipeline. A benefit is that we rely on open source models, therefore our pipeline can be extended to new datasets, however, it is still expensive. Furthermore, extensive post-processing, which is tailored to the LLM, is required along with manual inspection to ensure high-quality user preferences. Using smaller LLMs may affect the quality of the generated preferences and in turn affect performance of Mender. Finally, we rely on the presence of user reviews, which limits the applicability of our preference approximation to certain datasets.

## 6 Broader Impact

Preference discerning enables dynamic steering of recommendation models based on user preferences without the need for re-training, potentially avoiding echo chambers by explicitly stating what content should be recommended. Furthermore, preference discerning can positively affect the user experience as it allows interaction with the recommendation system, fostering trust and transparency. However, potential risks include amplifying biases if preferences reflect societal prejudices or leading to over-personalization.

## 7 Conclusion

Current sequential recommendation models are limited in their personalization as they *implicitly* model user preferences. We propose a new paradigm, namely preference discerning, in which the sequential recommendation system is *explicitly* conditioned on user preferences represented in natural language. To evaluate preference discerning capabilities, we present a benchmark that is specifically designed to evaluate the ability of sequential recommendation models to discern textual preferences along five different axes. We also propose a novel generative retrieval model, Mender, which represents items at different levels of abstraction, namely semantic IDs and natural language. Our experimental results show that Mender outperforms the state-of-the-art models on our benchmark and can adapt to unseen preferences without any retraining. Our contributions pave the way for a new class of generative retrieval models with the ability to dynamically adapt to user preferences provided in their context.

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

# Supplementary Material

**Fabian Paischer**
*ELLIS Unit, LIT AI Lab, Institute for Machine Learning, JKU Linz, Austria*
*AI at Meta*

**Liu Yang**
*University of Wisconsin-Madison*
*AI at Meta*

**Linfeng Liu, Shuai Shao, Kaveh Hassani, Jiacheng Li, Ricky Chen, Zhang Gabriel Li, Xiaoli Gao, Wei Shao, Xue Feng, Nima Noorshams, Sem Park, Bo Long, Hamid Eghbalzadeh**[†]
*heghbalz@meta.com*
*AI at Meta*

**Reviewed on OpenReview:** *https://openreview.net/forum?id=74mrOdhvvT*

## Contents

## A   Generative Retrieval via semantic IDs

We provide an open source implementation of all baselines used in this work, including TIGER (Rajput et al., 2023). To facilitate reproducibility of the results reported in Rajput et al. (2023), we elaborate on the implementation details as follows. The training of TIGER consists of two stages: (i) training the residual quantizer (RQ-VAE) to obtain semantic IDs, and (ii) training the generative retrieval model.

### A.1   RQ-VAE

Training the RQ-VAE involves two essential steps: (i) constructing an item embedding, and (ii) optimizing the model through residual quantization.

**Item embedding** For item embedding, we utilize the Sentence-T5 model (Ni et al., 2022), which is publicly available on the Hugging Face Hub (Wolf et al., 2020). We explored various sources of information to represent items and found that the optimal approach varies between datasets. For the Beauty and Sports datasets, using item descriptions led to suboptimal results due to the high noise levels present in these descriptions. In contrast, item descriptions proved beneficial for the Toys dataset. Additionally, we leveraged other item attributes, including title, price, brand, and categories. For the Stream dataset, we utilized a broader set of attributes: title, genre, specs, tags, price, publisher, and sentiment.

**Training** By default, we standardize the item embeddings, as this helps prevent collapse during RQ-VAE training. For training the RQ-VAE, we found that architectural changes are crucial to increase codebook coverage. Specifically, residual connections and weight decay are essential for maintaining a good separation. Our trained RQ-VAE's consistently attain a codebook coverage of more than 95%. Our encoder architecture consists of four hidden layers with sizes 768, 512, 256, and 128, respectively. Each layer includes layer normalization (Ba et al., 2016), ReLU activation, and dropout (Hinton et al., 2012). The decoder follows the same architecture but in reverse order, where the sum of residuals obtained via the quantization procedure is up-projected to the original dimension of 768. Following Rajput et al. (2023), we use a three-level residual quantization scheme with 256 codebooks each. We also experimented with EMA updates and resetting unused codebook entries, as in Lee et al. (2022), but did not observe any significant improvements. To evaluate the performance of our trained RQ-VAEs, we rely on metrics such as reconstruction error, codebook coverage, and downstream task performance.

## A.2 Transformer

Following Rajput et al. (2023) we instantiate the generative model with the T5 architecture (Raffel et al., 2020). Next, we investigate the design choices that underlie this approach, as introduced by Rajput et al. (2023), and discuss their utility.

**Training sequences** To construct the training sequences, Rajput et al. (2023) limit the number of items in a user sequence to at most 20. This can be implemented by taking the first, the last, or all items within a sliding window of up to 20 items. We experimented with each of these approaches and found that using the most recent 20 items in a user sequence generally yields improved performance. Unlike prior sequential recommendation systems, which require at least one item in a sequence to predict the next item (Kang & McAuley, 2018; Zhou et al., 2020), TIGER uses a user embedding trained alongside the item embeddings. Therefore, we typically use the first item in a sequence for training as well.

**Model architecture.**

**Decoding** Another crucial aspect of the generative retrieval pipeline is the decoding process. As noted in Rajput et al. (2023), the generation of valid semantic IDs is not guaranteed. To mitigate this issue, we track the number of invalid semantic IDs produced during decoding. We find that this number is typically quite low. Nevertheless, to further improve the accuracy of our retrieval results, we employ filtering to remove invalid IDs and increase the beam size to be 30, which is larger than the final retrieval set.

## A.3 Reproduced results

In Table 2, we compare the results of our reproduced version of TIGER with those reported in Rajput et al. (2023). Our results closely match those reported in Rajput et al. (2023) for the Sports and Beauty datasets, but we observe a significant gap on the Toys dataset. In particular, our trained models achieve substantially higher Recall@10 scores on the Beauty dataset. Furthermore, we find that the disparity is more pronounced for NDCG than for Recall, suggesting that while the retrieved candidate items are similar, our models' ranking performance is slightly worse.

## A.4 Additional findings

Beyond the experiments discussed above, we conducted further investigations into the TIGER framework, yielding the following key insights.

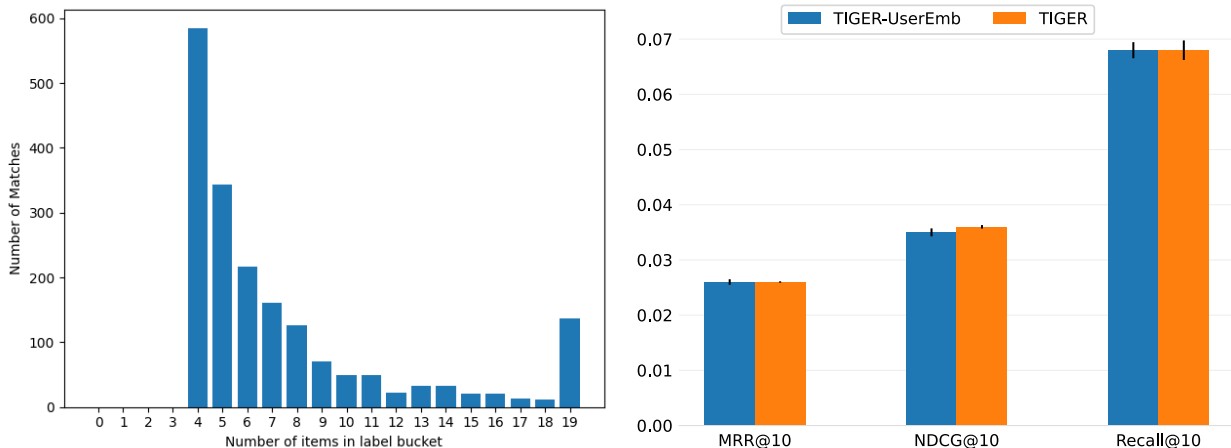

Figure 7: **Left:** Number of correctly retrieved test items for TIGER on the Beauty subset of the Amazon review dataset. **Right:** Performance comparison of TIGER with user embedding (TIGER-UserEmb) and without user embedding (TIGER) on the Beauty dataset.

Table 2: Reproduced results for our open-source implementation of TIGER (Rajput et al., 2023)

| Methods | Sports and Outdoors | | | | Beauty | | | | Toys and Games | | | |
|---|---|---|---|---|---|---|---|---|---|---|---|---|
| | Recall @5 | NDCG @5 | Recall @10 | NDCG @10 | Recall @5 | NDCG @5 | Recall @10 | NDCG @10 | Recall @5 | NDCG @5 | Recall @10 | NDCG @10 |
| P5 Geng et al. (2022) | 0.0061 | 0.0041 | 0.0095 | 0.0052 | 0.0163 | 0.0107 | 0.0254 | 0.0136 | 0.0070 | 0.0050 | 0.0121 | 0.0066 |
| Caser Tang & Wang (2018) | 0.0116 | 0.0072 | 0.0194 | 0.0097 | 0.0205 | 0.0131 | 0.0347 | 0.0176 | 0.0166 | 0.0107 | 0.0270 | 0.0141 |
| HGN Ma et al. (2019) | 0.0189 | 0.0120 | 0.0313 | 0.0159 | 0.0325 | 0.0206 | 0.0512 | 0.0266 | 0.0321 | 0.0221 | 0.0497 | 0.0277 |
| GRU4Rec Hidasi et al. (2016) | 0.0129 | 0.0086 | 0.0204 | 0.0110 | 0.0164 | 0.0099 | 0.0283 | 0.0137 | 0.0097 | 0.0059 | 0.0176 | 0.0084 |
| BERT4Rec Sun et al. (2019) | 0.0115 | 0.0075 | 0.0191 | 0.0099 | 0.0203 | 0.0124 | 0.0347 | 0.0170 | 0.0116 | 0.0071 | 0.0203 | 0.0099 |
| FDSA Zhang et al. (2019b) | 0.0182 | 0.0122 | 0.0288 | 0.0156 | 0.0267 | 0.0163 | 0.0407 | 0.0208 | 0.0228 | 0.0140 | 0.0381 | 0.0189 |
| SASRec Kang & McAuley (2018) | 0.0233 | 0.0154 | 0.0350 | 0.0192 | 0.0387 | 0.0249 | 0.0605 | 0.0318 | 0.0463 | 0.0306 | 0.0675 | 0.0374 |
| S$^3$-Rec Zhou et al. (2020) | 0.0251 | 0.0161 | 0.0385 | 0.0204 | 0.0387 | 0.0244 | 0.0647 | 0.0327 | 0.0443 | 0.0294 | 0.0700 | 0.0376 |
| **TIGER(Rajput et al., 2023)** | 0.0264 | 0.0181 | 0.0400 | 0.0225 | 0.0454 | 0.0321 | 0.0648 | 0.0384 | 0.0521 | 0.0371 | 0.0712 | 0.0432 |
| **TIGER (Ours)** | 0.0249 | 0.0158 | 0.0377 | 0.0199 | 0.0431 | 0.0275 | 0.0681 | 0.0356 | 0.0375 | 0.0238 | 0.0600 | 0.0311 |

- TIGER exhibits superior performance on shorter sequences, as shown in Fig. 7 (left).
- The inclusion of user embeddings in TIGER does not yield any significant benefits to downstream performance, as illustrated in Fig. 7 (right).
- Representing the interaction history in natural language leads to improved ranking performance, as demonstrated in Fig. 8.

**TIGER Works Better on Shorter Sequences.** As shown in Fig. 7 (left), TIGER performs significantly better on shorter sequences than on longer ones. The x-axis represents the number of items per test sequence, which is at least 4 due to the 5-core user and item filtering applied. Further, the maximum number of items per sequence is capped at 19, as we limit the maximum sequences length to 20, following (Rajput et al., 2023). This results in a maximum sequence length of 19 items, where the task is to predict the 20th item. The y-axis shows the number of matches. Notably, TIGER's performance is substantially better on shorter sequences than on longer ones. However, the number of matches increases again for the longest sequences, although it remains considerably lower than for shorter sequences.

**User Embedding.** Rajput et al. (2023) employ a user embedding selected based on hashing. However, it is unclear whether this approach offers any advantages, as the number of user embeddings suggested by Rajput et al. (2023) often results in numerous collisions in practice. To investigate this, we conduct an experiment

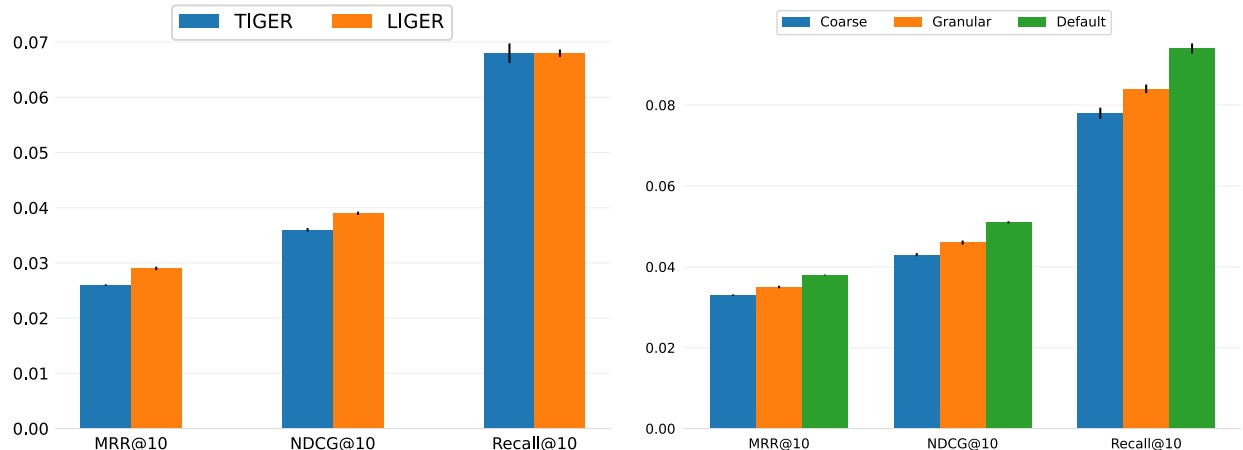

Figure 8: **Left:** Performance comparison between TIGER and LIGER on the Beauty subset of the Amazon review dataset. Both models predict semantic IDs, but differ in their input representation: LIGER encodes past items as natural language descriptions, while TIGER represents them as semantic IDs. **Right:**

in which we remove the user embedding entirely. As shown in Fig. 7 (middle), we do not observe a significant drop in performance. This suggests that user embedding does not provide any notable benefits.

**History Compression via Natural Language.** We conduct an additional study in which we represent the past interaction history in text and initialize the TIGER encoder with a small FLAN-T5 encoder (Chung et al., 2024). This approach is reminiscent of history compression via language models (Paischer et al., 2022, HELM). We refer to this variant as LIGER (Language-TIGER), and compare its performance with the baseline TIGER in Fig. 8, left. The results show that while there is no significant difference in Recall, LIGER yields notable improvement in NDCG metrics. This suggests that compressing interaction history using natural language generally enhances the model's ranking capabilities.

# B    Datasets

We consider two publicly available datasets for sequential recommendation: Amazon review dataset (Ni et al., 2019) and Steam (Kang & McAuley, 2018). To preprocess these datasets, we apply 5-core filtering criterion, removing users with fewer than five interactions and items that appear less than five times. The statistics of the resulting dataset are presented in Table 3. Due to computational constraints, we sub-sample the Steam dataset to reduce the number of user preferences generated during the preference approximation pipeline.

We also visualize the item distribution in Fig. 9, which shows that the three Amazon datasets follow approximately the same item distribution, while for Steam the distribution differs significantly. In particular, on the Steam dataset the number of items is in the same range as for the Amazon datasets; however, the number of users is much larger, as well as the average number of actions per user. As can be observed from the item distribution, there is a small fraction of items that is overrepresented.

# C    Preference generation

In this section, we provide details on the prompting scheme used to generate user preferences from item reviews using `LLaMA-3-70B-Instruct`. We provide reviews along with item-specific information to the LLM and prompt it to generate a set of five user preferences (see Fig. 10). Below we present an example prompt and response for a user in the Beauty subset of the Amazon reviews dataset.

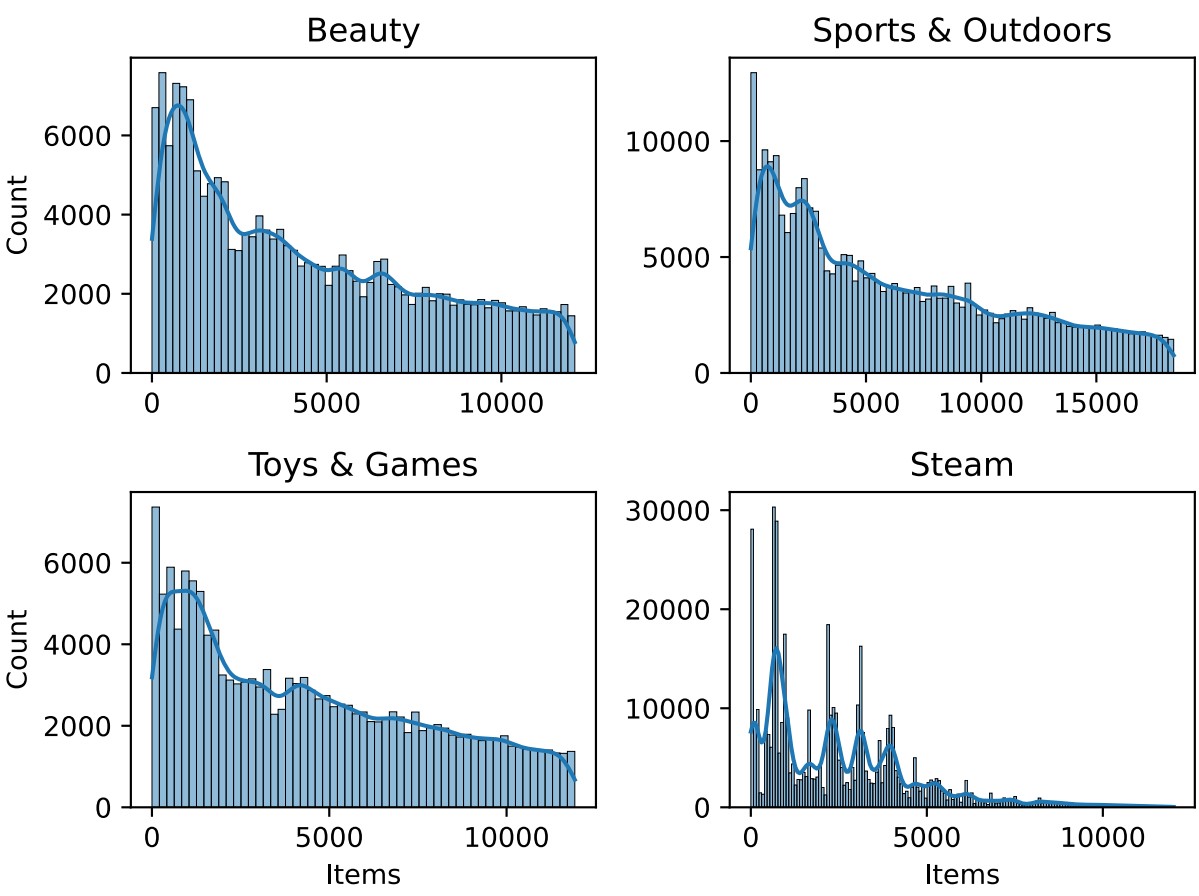

Figure 9: Data distribution of the Amazon and Steam datasets.

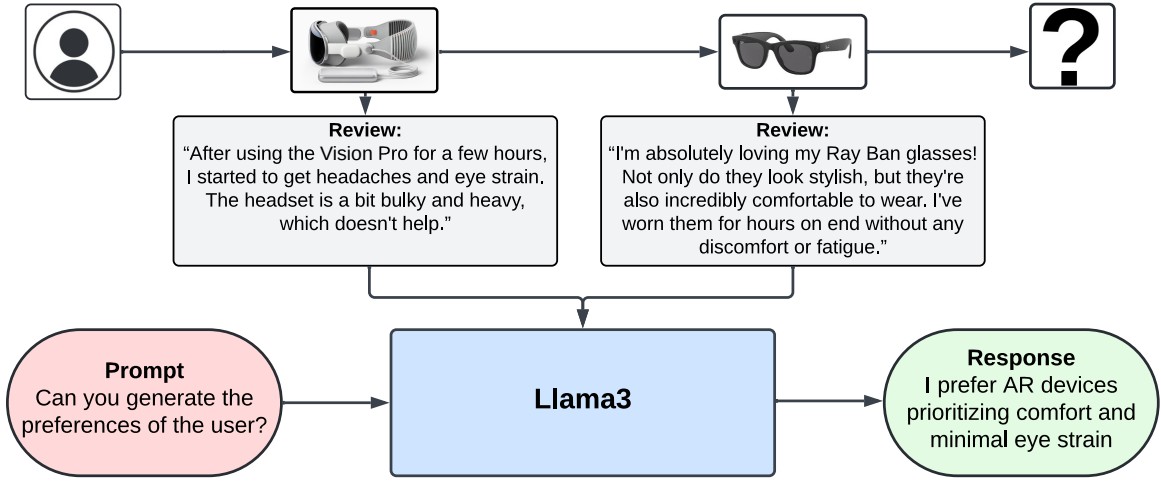

Figure 10: Schematic illustration of our preference generation pipeline. A user's reviews for items, combined with item information, are input into Llama3 as a prompt to infer the user's preferences.

Table 3: Dataset statistics after user 5-core and item 5-core preprocessing. Asterisk denotes datasets are subsets of the Amazon review dataset.

| Dataset | #users | #items | avg. actions /user | avg. actions /item | #actions |
|---|---|---|---|---|---|
| *Beauty\** | 22,363 | 12,101 | 8.8764 | 16.403 | 198,502 |
| *Toys and Games\** | 19,412 | 11,924 | 8.6337 | 14.0554 | 167,597 |
| *Sports and Outdoors\** | 35,598 | 18,357 | 8.3245 | 16.1430 | 296,337 |
| *Yelp* | 19,855 | 14,540 | 10.4279 | 14.2387 | 207,045 |
| *Steam* | 47,761 | 10,403 | 12.554 | 54.6549 | 599,620 |

Table 4: Statistics for generated preferences for the different datasets. For pos/neg and fine/coarse we show number of samples in the format train/val/test split.

| Benchmark | #preferences | #positive | #negative | pos/neg | fine/coarse |
|---|---|---|---|---|---|
| *Beauty* | 992,510 | 708,706 | 283,804 | 17,811/3,671/3,716 | 24,114/16,702/15,956 |
| *Toys and Games* | 837,985 | 645,696 | 192,289 | 11,513/2,342/2,508 | 23,730/15,968/14,950 |
| *Sports and Outdoors* | 1,481,685 | 1,075,679 | 406,006 | 21,402/4,275/4,293 | 36,552/25,728/25,188 |
| *Steam* | 2,026,225 | 1,495,931 | 530,294 | 31,505/7,968/8,493 | 19,550/10,678/10,626 |

---

**Instruction:**
Here is a list of items a user bought along with their respective reviews in json format: `{ }`. Your task is to generate a list of up to five search instructions that reflect the user's preferences based on their reviews. Be specific about what the user likes, does not like, and should be avoided. Do not mention brands or certain products. Return a json file containing the search instructions with the key 'instructions'. Keep the instructions simple, short and concise, and do NOT include comments on delivery time or pricing.

**Parsed response:**
```
['Search for nail polish with shimmer finish', 'Look for products with vibrant, bold colors',
 'Avoid products that require base coat for optimal results', 'Prioritize products with high-quality,
 long-lasting formula', 'Opt for products with easy, smooth application']
```

---

After generation, we apply an exhaustive postprocessing step to ensure that every user-item pair is associated with exactly five user preferences. In Table 4 we show the statistics after our preference generation pipeline for the different datasets.

**Granularity of preferences.** We also investigate whether the granularity of user preferences affects the model's ability to adhere to them. We experiment with various prompts to elicit preferences at different levels of granularity, ranging from coarser, higher-level user preferences to fine-grained preferences that include detailed descriptions of products and brands. To obtain more abstract user preferences, we slightly modify the prompt as follows. We show an example for the same user below.

> **Instruction:**
> Here is a list of items a user bought along with their respective reviews in json format: *{ }*. Your task is to generate a list of up to five search instructions that summarizes the user's high-level preferences based on their reviews. Be specific on what the user does not like and should be avoided. Do not mention brands or certain products. Return a json file containing the search instructions with the key 'instructions'. Keep the instructions simple, short and concise, and do NOT include comments on delivery time or pricing.
>
> **Parsed response:**
> *['Look for products with vibrant colors', 'Prioritize products with high-quality finishes', 'Opt for products that are easy to apply', 'Consider products that are suitable for multiple uses (e.g., toes and fingers)', 'Avoid products that require additional base coats']*

Finally, to obtain more fine-grained user preferences, we slightly modify the prompt again to permit the inclusion of specific brands and item names in the LLama response. As shown below, the LLama indeed mentions brands when given the explicit instruction that mentioning item names or brands is acceptable.

> **Instruction:**
> Here is a list of items a user bought along with their respective reviews in json format: *{ }*. Your task is to generate a list of up to five search instructions that reflect the user's preferences based on their reviews. Be specific about what the user likes, does not like, and should be avoided. It is okay to mention brands or certain products. Return a json file containing the search instructions with the key 'instructions'. Keep the instructions simple, short and concise, and do NOT include comments on delivery time or pricing.
>
> **Parsed response:**
> *["Search for nail polishes with similar shimmering effects to OPI's Simmer and Shimmer", 'Prioritize products with vibrant, long-lasting colors that look great on toes', "Look for nail polish brands that offer a wide range of colors, similar to OPI's Burlesque collection", 'Avoid products that require a base coat for optimal results', 'Opt for nail polishes with a smooth, easy-to-apply formula']*

In practice, we found that varying the granularity usually leads to worse performance (see Fig. 8, right). This suggests that the user preferences obtained through our default prompting scheme contain sufficient information to effectively steer the model.

## C.1 Reviews to properties

In addition to the title and description of the items, the collective reviews of items offer a wealth of information about their properties. However, with hundreds of users potentially reviewing a single item, each contributing multiple paragraphs, extracting relevant information can be challenging. To improve the signal-to-noise ratio, we prompt `LLaMA-3-70B-Instruct` to extract and condense the properties of the item from the corpus of user reviews. Specifically, we instruct the LLM to provide concise objective descriptions, excluding subjective opinions. In the following, we illustrate this approach with an example of extracting properties from reviews of a froggy pool in the Amazon Toys and Games dataset.

> **Instruction:**
> Your task is to summarize the following reviews of an item into a list of item properties using keywords and phrases: *{ }* . Keep your response short and concise. Only focus on objective properties of the item. Do NOT include subjective opinions or emotions. Do NOT include comments on price or delivery time. Return your response as a python list with at most 10 entries that accurately reflect the properties of the item.
> **Parsed response:**
> `Inflatable pool with a frog head design, Rainbow sunshade provides shade for the entire pool, inflatable base consists of multiple air cushions, pool holds 12 gallons of water, Maximum fill line is 4 inches, Inflated pool area has a 28 inch diameter, Three separate air valves for inflation, Easy to set up and take down, Suitable for children aged 1-3 years, Cushioned inflatable base for comfort.`
> **Original review:**
> `This froggy pool is great for your young children. It is 45" x 39" x 27" and very comfortable and safe for your child.`

The parsed response above demonstrates that LLama3 is capable of providing a comprehensive list of characteristics about toys in the Amazon Toys and Games dataset. We observed modest performance gains when providing summarized item properties along with item titles and preferences as input to our language-conditioned generative retrieval model. Since we were limited by the number of prompts we could issue, we did not compile the item properties for all datasets. Nevertheless, we provide the prompts to facilitate future research in this direction.

## D   Benchmark design

In this section, we provide additional details on the creation of the various components of our benchmark.

### D.1   Sentiment Following

The sentiment understanding benchmark is based on preference-item pairs and utilizes a matching mechanism to identify items that triggered negative reviews. This is implemented using a pre-trained sentiment classification model from Hartmann et al. (2023) to classify reviews. To identify preferences, we employ a rule-based approach, as we observed that preferences can be both positive and negative simultaneously (e.g., a preference may specify liking certain items while avoiding others). Furthermore, we noticed that negative preferences consistently follow a specific pattern, starting with either "*Avoid*", "*Exclude*", or "*No*". To reduce misclassifications, we consider preferences beginning with these words as negative. If only one item in a user sequence received a negative review, we pair the negative preference with that item. Otherwise, we use a matching mechanism in the Sentence-T5 space, where we match a negative preference to the item whose review is closest in terms of cosine similarity. An example of the negative matching pipeline is illustrated in Fig. 11 . This yields a set of negative preference-item pairs, enabling us to assess whether the model can recognize negative sentiment and respond accordingly. To obtain positive pairs of preferences-items, we iterate over all negative pairs and invert the gathered preferences. Since negative instructions always start with "*Avoid*", "*Exclude*", or "*No*", we simply replace these words with "*Find*" or "*Search for*" to invert them. This results in two sets: one that contains negative preferences paired with items and another containing positive preferences paired with the same items. Finally, we assess whether the model can successfully avoid certain items while actively retrieving others.

### D.2   Preference Steering

In the preference steering scenario, we consider two distinct scenarios: *fine-grained* and *coarse-grained* preference steering. The former assesses whether the model can retrieve an item very similar to the ground truth by modifying the user preference. In contrast, the latter evaluates whether the model can retrieve a distinctly different item by changing the user preference accordingly. We identify a very similar item by the maximal cosine similarity in a pre-trained Sentence-T5 embedding space. In contrast, we retrieve a very distinct item by the lowest cosine similarity to the ground-truth item. Subsequently, we match the retrieved

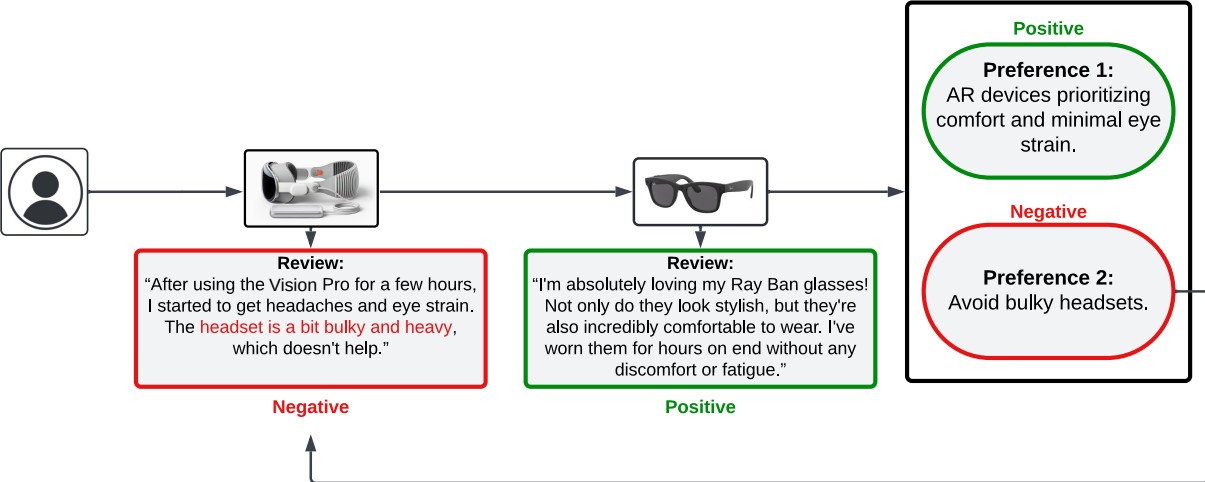

Figure 11: Schematic illustration of our pipeline to identify the reviews that triggered negative user preferences. The reviews of different items guided the LLM to generate two distinct user preferences. We perform sentiment classification on both user preferences and reviews, followed by a matching step in Sentence-T5 space to determine which negative review led to a negative user preference.

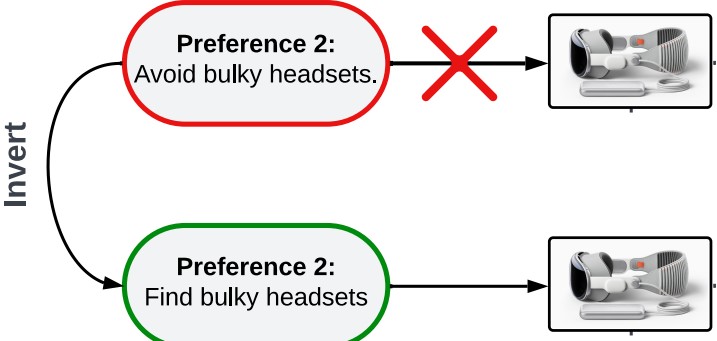

Figure 12: Positive and negative preference-item pairs obtained after matching negative preferences to items that received a negative review. We apply a rule-based inversion to generate the corresponding positive pair.

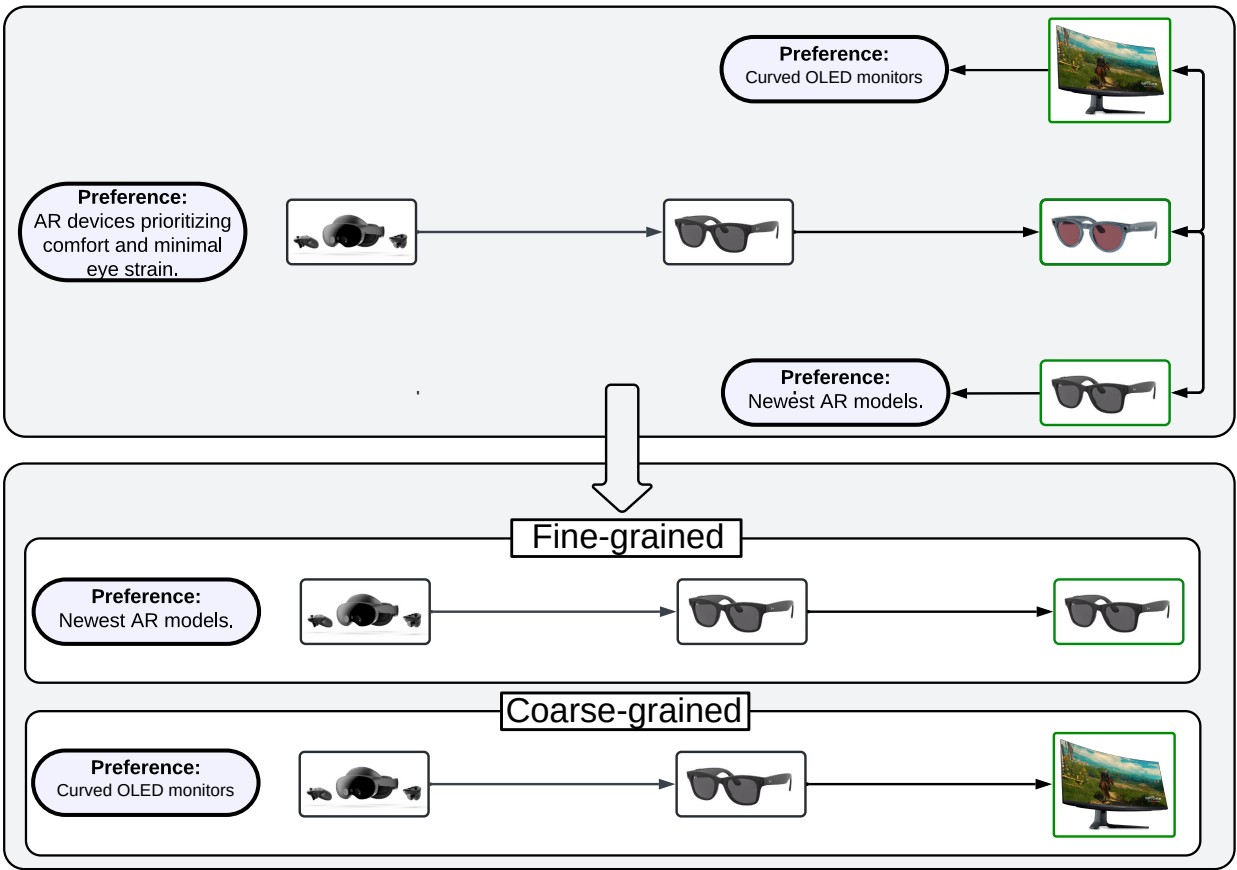

Figure 13: Schematic illustration of our pipeline for constructing fine- and coarse-grained preference steering. We search for very similar and dissimilar items to the ground truth item of each original item sequence and match them to user preferences (top). Then, we obtain two new sequences by exchanging the original preference with each user preferences and associated new ground truth item.

items to new user preferences, again via cosine similarity. We show a visual illustration of this procedure in Fig. 13. Finally, we ensure that there is no overlap between our compiled training, validation, and test split by controlling for the matched preferences, i.e. if a user preference was already matched to a retrieved item, we associate the current item with the next most similar or distinct preference. This results in unique (preference, item) tuples for every dataset split.

# E    Additional results

We provide complementary results for our ablation studies on the data mixture. In Table 5 we report Recall@5, Recall@10, NDCG@5 and NDCG@10 for the different versions of Mender that are trained on different data mixes. Furthermore, we provide results for training on the Steam dataset with different data mixtures in Fig. 14 to highlight that fine- and coarse-grained steering, as well as sentiment following capabilities can be obtained on this dataset as well.

In addition, we report standard deviations of our results in Table 1 in Table 6 with the higher values colored red. The small standard deviation indicates that the improvements reported in Mender are statistically significant.

To assess the efficiency of our Mender variants, we compare the time required for training and inference as well as their performance. Furthermore, we add a comparison to SASRec (Kang & McAuley, 2018), which is a traditional sequential recommendation baseline. We present our results in Table 7 for the four datasets.

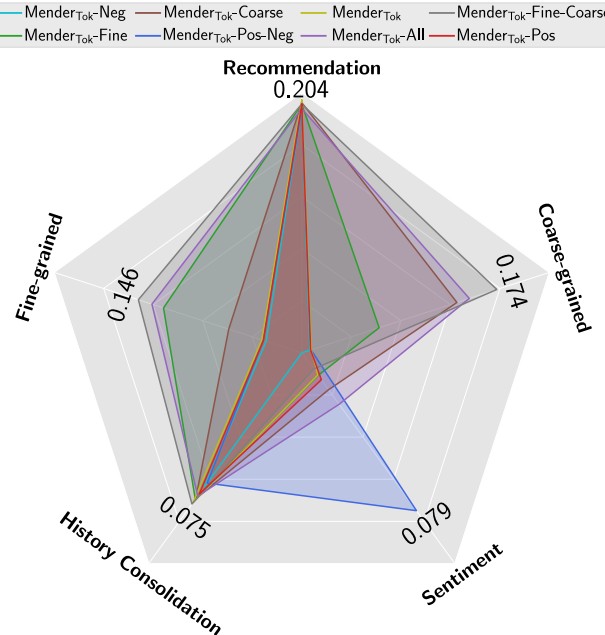

Figure 14: Recall@10 for Mender$_\text{Tok}$ trained on different datasplits on the Steam dataset, evaluated under various schemes: *Recommendation*, *Sentiment following*, *Preference steering*, *Preference consolidation*, and *History consolidation.*

In addition, we conduct an experiment to demonstrate that training on all five generated user preferences leads to detrimental performance. As mentioned in Section 3.5, each training sequence contains a single user preference that is matched to the target item in a pre-trained SentenceT5 space. To verify that this is the best training strategy, we compare Mender$_\text{Tok}$ trained on these sequences to the setup where Mender$_\text{Tok}$ receives all five user preferences along with the interaction history (Mender$_\text{Tok}$-AllPrefs), that is, the training sequences are structured as $\left[p_{u_1}^{T_u-1}, \ldots, p_{u_5}^{T_u-1}, i_1, \ldots, i_{T_u-1}\right]$. We report our results in Table 8. They verify that training on sequences $\left[p_u^{T_u-1}, \ldots, i_1, \ldots, i_{T_u-1}\right]$ where $p_u^{T_u-1}$ is matched to the ground truth item $i_{T_u-1}$ attains significantly better results than training on providing all preferences in the sequence.

Finally, we conduct an experiment where we exchange the language encoder of Mender$_\text{Tok}$ with a larger variant. By default, all experiments use the FLAN-T5-Small model (Chung et al., 2024). In Table 9 we provide results for a comparison to the XXL variant. We observe that drastic improvements can be obtained on certain datasets usually for tasks such as fine-grained steering, coarse-grained steering, or sentiment following. This provides evidence that, for the more language-intensive tasks, it is beneficial to scale the language encoder. However, on Beauty, Toys and Games and Steam, there are some discrepancies, which are mainly due to the fact that we fine-tune the small variant, but not the large one as this lead to improved performance. Due to computational requirements, we do not fine-tune the XXL encoder. Impressively, the XXL variants improve performance on fine- and coarse-grained steering on the Toys and Games and the Steam datasets, even though we compare to the fine-tuned small variant. This provides compelling evidence that more capable models can lead to drastic improvements on the different performance axes. Finally, the XXL variant leads to a drastic improvement on the history consolidation task on Steam, indicating that a better language understanding is required to tackle this task on the Steam dataset.

## F   Manual Inspection of Preferences

Our aim is to verify that the user preferences generated by the LLM accurately approximate the real user preferences. To this end, we conduct a manual confirmation of the preferences to answer the following questions:

1. Are the generated user preferences informed by the user's past interaction history?
2. Do the generated preferences accurately approximate the user's preferences?
3. Is the matched preference related to the target item?
4. Given that a user preference accurately approximates the user's preferences, is it related to the target item?

In total, we manually inspected 440 recommendation scenarios, which is equivalent to 2200 preferences that were judged. Each scenario consists of 20 randomly sampled recommendations of one of the Beauty, Toys and Games, Sports and Outdoors, or Steam datasets. In one of such scenario, we first show the past interaction history of a random user along with their reviews. Then, the generated user preferences are displayed along with the one user preference that was matched to the ground-truth item, i.e. the next item in the sequence. Finally, we also display the ground truth item with the same information as the recommendation system would receive it. For each scenario, we answer all three aforementioned questions and provide one of three possible answers, namely (1) yes, (2) no, or (3) lack of information to tell. We now iterate over all the questions and present the corresponding findings.

**Are the generated user preferences informed by the user's past interaction history?** The objective of introducing this question was to quantify how much of the generated preferences was actually represented in the interaction history and what amount has been *hallucinated*. We report the results of this first question in Fig. 15. The majority of generated user preferences are well informed by the user's interaction history across datasets. We found that the model occasionally generated rather generic preferences, for example, "Avoid harsh chemicals" on the Beauty dataset, although there was no mention of "harsh chemicals" in the reviews. Such preferences are rather generic and do not convey much information about a user's preference. Furthermore, there was a lack of information in some scenarios to answer the question. This can be traced back to the fact that we intentionally did not provide item descriptions, as these often contain a substantial amount of noise. As this information is hidden, we believe that it caused the small fraction of preferences that were rated as *lack of info*. Thus, we can conclude that the generated user preferences for the most part were informed by reviews or item-specific info, however, there is still a non-negligible amount of user preferences that can be considered *hallucinated*.

**Do the generated preferences accurately approximate the user's preferences?** The purpose of this question is to quantify whether user preferences are correctly approximated. This question is crucial because it sits at the core of evaluating the quality of preferences. We report the result in Fig. 16. Again, we find that, for the most part, the preferences accurately reflect the user's preferences. The answer *lack of info* means that there is not enough information to capture the user's preferences, which is the case if very little detail is given in the reviews or they are missing entirely. Fortunately, this case is underrepresented. Overall, we can conclude that the approximation of user preferences via our preference approximation strategy yields high-quality preferences that accurately reflect the user's preferences.

**Is the matched preference related to the target item?** After we have established the quality of the preferences, it is imperative to also evaluate our matching of preferences to target items for preference-based recommendation. The reason we conduct this matching is to provide the model with a useful training signal. This is imperative as we observed that simply using all preferences for training leads to detrimental performance (see Table 8). We report the results for this question in Fig. 17. Interestingly, the fraction of correctly matched preferences is significantly lower compared to the number of correctly generated preferences. The reasons for this can be two fold, (i) it can be that the target item is entirely unrelated to the past interaction history, or (ii), the matching mechanism is suboptimal. The former case reflects the inherent aleatoric uncertainty of the sequential recommendation task, as oftentimes the target item is simply not related to previously acquired purchases. This shortcoming cannot be alleviated. However, the latter can be tackled by potentially more expressive embedding models or LLMs that can be used to match preferences to the target item. Finally, the *lack of info* category represents cases where the information about the target item is too little, i.e., no description or item title is given. Overall, we can conclude that even though we demonstrated significant performance gains resulting from training on the matched preferences, it could likely be further improved by improving the matching of preferences to items.

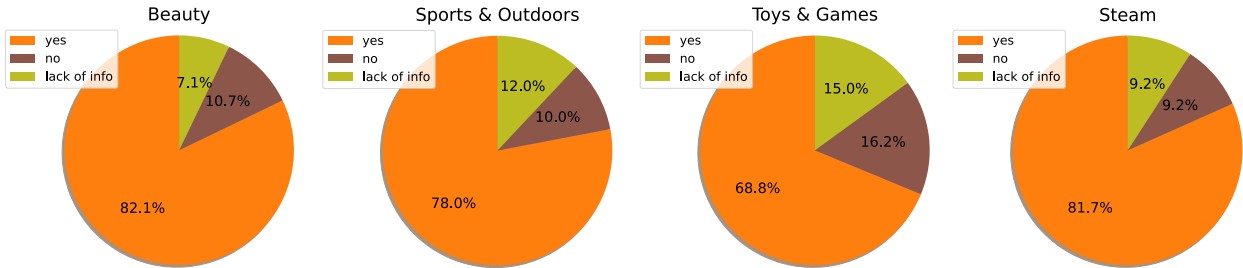

Figure 15: Manual inspection results for the question "Are the generated user preferences informed by the user's past interaction history?" for the four different datasets used for approximating user preferences.

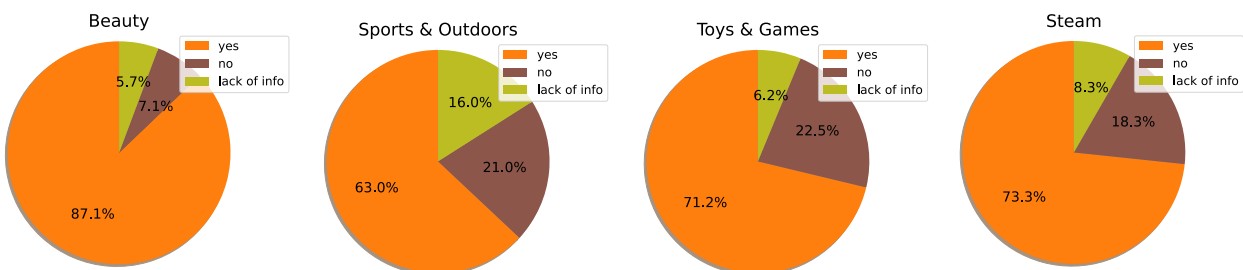

Figure 16: Manual inspection results for the question "Do the generated preferences accurately approximate the user's preferences?" for the four different datasets used for approximating user preferences.

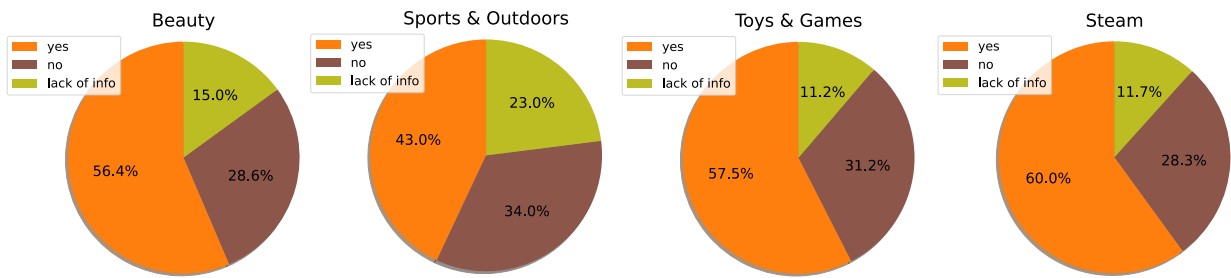

Figure 17: Manual inspection results for the question "Is the matched preference related to the target item?" for the four different datasets used for approximating user preferences.

**Given that a user preference accurately approximates the user's preferences, is it related to the target item?** We can obtain an estimate on the underlying aleatoric uncertainty of the task by evaluating whether accurate preferences are related to the target item. In particular, we consider the cases where Q2 was answered yes and visualize the three categories for Q3 (see Fig. 18). In other words, we look at correctly approximated preferences and ask what fraction of them is related to the target item. If Q2 is answered with *yes*, then we expect the matching to perform well if there is a semantic relation to the target item. However, if there is still no relation to the target item, that is, Q3 is answered with *no*, then we can infer that this is due to the inherent uncertainty of the task. Interestingly, 50-70% of the correctly approximated preferences are related to the target item. This provides us with an empirical upper bound on the maximum performance that can be obtained on the sequential recommendation task, i.e. the maximum Recall that can be obtained is in the range of 0.5-0.7, depending on the dataset.

Table 5: Performance for different versions of Mender trained on different data mixtures for all evaluation axes on the Beauty and Steam datasets. We report average performance across three random seeds.

| Methods | Beauty | | | | Steam | | | |
|---|---|---|---|---|---|---|---|---|
| | Recall @5 | NDCG @5 | Recall @10 | NDCG @10 | Recall @5 | NDCG @5 | Recall @10 | NDCG @10 |
| Recommendation | | | | | | | | |
| $Mender_{Tok}$ | 0.0605 | 0.0401 | 0.0937 | 0.0508 | 0.1682 | 0.1441 | 0.2037 | 0.1555 |
| $Mender_{Tok}$-Pos | 0.0553 | 0.0371 | 0.0840 | 0.0463 | 0.1667 | 0.1429 | 0.2004 | 0.1538 |
| $Mender_{Tok}$-Neg | 0.0598 | 0.0394 | 0.0917 | 0.0497 | 0.1646 | 0.1410 | 0.1983 | 0.1519 |
| $Mender_{Tok}$-Pos-Neg | 0.0491 | 0.0321 | 0.0778 | 0.0413 | 0.1647 | 0.1416 | 0.1979 | 0.1523 |
| $Mender_{Tok}$-Fine | 0.0591 | 0.0383 | 0.0918 | 0.0487 | 0.1667 | 0.1428 | 0.2005 | 0.1538 |
| $Mender_{Tok}$-Coarse | 0.0601 | 0.0392 | 0.0924 | 0.0496 | 0.1682 | 0.1440 | 0.2018 | 0.1549 |
| $Mender_{Tok}$-Fine-Coarse | 0.0570 | 0.0366 | 0.0893 | 0.0470 | 0.1663 | 0.1424 | 0.2007 | 0.1535 |
| $Mender_{Tok}$-All | 0.0529 | 0.0337 | 0.0838 | 0.0436 | 0.1634 | 0.1400 | 0.1969 | 0.1508 |
| Fine-grained steering | | | | | | | | |
| $Mender_{Tok}$ | 0.0534 | 0.0344 | 0.0844 | 0.0444 | 0.0218 | 0.0137 | 0.0357 | 0.0182 |
| $Mender_{Tok}$-Pos | 0.0501 | 0.0321 | 0.0791 | 0.0414 | 0.0217 | 0.0137 | 0.0343 | 0.0177 |
| $Mender_{Tok}$-Neg | 0.0500 | 0.0323 | 0.0803 | 0.0420 | 0.0196 | 0.0124 | 0.0318 | 0.0163 |
| $Mender_{Tok}$-Pos-Neg | 0.0513 | 0.0333 | 0.0791 | 0.0423 | 0.0211 | 0.0131 | 0.0344 | 0.0173 |
| $Mender_{Tok}$-Fine | 0.2476 | 0.1680 | 0.3475 | 0.2002 | 0.0829 | 0.0538 | 0.1234 | 0.0668 |
| $Mender_{Tok}$-Coarse | 0.1483 | 0.0981 | 0.2212 | 0.1215 | 0.0395 | 0.0244 | 0.0652 | 0.0327 |
| $Mender_{Tok}$-Fine-Coarse | 0.2781 | 0.1885 | 0.3861 | 0.2234 | 0.0985 | 0.0643 | 0.1459 | 0.0795 |
| $Mender_{Tok}$-All | 0.2676 | 0.1802 | 0.3750 | 0.2148 | 0.0903 | 0.0601 | 0.1338 | 0.0741 |
| Coarse-grained steering | | | | | | | | |
| $Mender_{Tok}$ | 0.0094 | 0.0059 | 0.0161 | 0.0080 | 0.0045 | 0.0028 | 0.0085 | 0.0041 |
| $Mender_{Tok}$-Pos | 0.0098 | 0.0062 | 0.0163 | 0.0083 | 0.0047 | 0.0029 | 0.0079 | 0.0040 |
| $Mender_{Tok}$-Neg | 0.0063 | 0.0039 | 0.0117 | 0.0056 | 0.0041 | 0.0027 | 0.0072 | 0.0036 |
| $Mender_{Tok}$-Pos-Neg | 0.0095 | 0.0061 | 0.0169 | 0.0084 | 0.0050 | 0.0031 | 0.0083 | 0.0041 |
| $Mender_{Tok}$-Fine | 0.1005 | 0.0655 | 0.1494 | 0.0813 | 0.0272 | 0.0175 | 0.0691 | 0.0304 |
| $Mender_{Tok}$-Coarse | 0.3028 | 0.2631 | 0.3541 | 0.2797 | 0.0953 | 0.0485 | 0.1385 | 0.0624 |
| $Mender_{Tok}$-Fine-Coarse | 0.3525 | 0.2710 | 0.4413 | 0.2999 | 0.1403 | 0.1052 | 0.1741 | 0.1163 |
| $Mender_{Tok}$-All | 0.3294 | 0.2779 | 0.3885 | 0.2970 | 0.1063 | 0.0696 | 0.1495 | 0.0839 |
| Sentiment following | | | | | | | | |
| $Mender_{Tok}$ | 0.0043 | - | 0.0053 | - | 0.0084 | - | 0.0110 | - |
| $Mender_{Tok}$-Pos | 0.0113 | - | 0.0140 | - | 0.0123 | - | 0.0134 | - |
| $Mender_{Tok}$-Neg | 0.0000 | - | 0.0000 | - | 0.0000 | - | 0.0000 | - |
| $Mender_{Tok}$-Pos-Neg | 0.0268 | - | 0.0414 | - | 0.0637 | - | 0.0787 | - |
| $Mender_{Tok}$-Fine | 0.0046 | - | 0.0075 | - | 0.0080 | - | 0.0112 | - |
| $Mender_{Tok}$-Coarse | 0.0067 | - | 0.0089 | - | 0.0088 | - | 0.0184 | - |
| $Mender_{Tok}$-Fine-Coarse | 0.0057 | - | 0.0083 | - | 0.0053 | - | 0.0081 | - |
| $Mender_{Tok}$-All | 0.0440 | - | 0.0635 | - | 0.0184 | - | 0.0256 | - |
| History consolidation | | | | | | | | |
| $Mender_{Tok}$ | 0.0457 | 0.0304 | 0.0720 | 0.0388 | 0.0490 | 0.0317 | 0.0745 | 0.0399 |
| $Mender_{Tok}$-Pos | 0.0405 | 0.0272 | 0.0632 | 0.0344 | 0.0490 | 0.0331 | 0.0704 | 0.0400 |
| $Mender_{Tok}$-Neg | 0.0460 | 0.0301 | 0.0714 | 0.0383 | 0.0448 | 0.0288 | 0.0667 | 0.0359 |
| $Mender_{Tok}$-Pos-Neg | 0.0359 | 0.0233 | 0.0581 | 0.0305 | 0.0440 | 0.0293 | 0.0649 | 0.0360 |
| $Mender_{Tok}$-Fine | 0.0418 | 0.0270 | 0.0657 | 0.0346 | 0.0492 | 0.0333 | 0.0730 | 0.0410 |
| $Mender_{Tok}$-Coarse | 0.0436 | 0.0284 | 0.0682 | 0.0363 | 0.0495 | 0.0331 | 0.0728 | 0.0406 |
| $Mender_{Tok}$-Fine-Coarse | 0.0399 | 0.0254 | 0.0636 | 0.0331 | 0.0517 | 0.0355 | 0.0753 | 0.0430 |
| $Mender_{Tok}$-All | 0.0379 | 0.0236 | 0.0607 | 0.0309 | 0.0506 | 0.0349 | 0.0713 | 0.0416 |

Table 6: Standard deviation for all methods on all evaluation axes for all datasets trained on recommendation data across three random seeds.

| Methods | Sports and Outdoors | | | | Beauty | | | | Toys and Games | | | | Steam | | | |
|---|---|---|---|---|---|---|---|---|---|---|---|---|---|---|---|---|
| | Recall@5 | NDCG@5 | Recall@10 | NDCG@10 | Recall@5 | NDCG@5 | Recall@10 | NDCG@10 | Recall@5 | NDCG@5 | Recall@10 | NDCG@10 | Recall@5 | NDCG@5 | Recall@10 | NDCG@10 |
| *Recommendation* | | | | | | | | | | | | | | | | |
| TIGER | 0.0009 | 0.0006 | 0.0006 | 0.0005 | 0.0010 | 0.0009 | 0.0012 | 0.0009 | 0.0008 | 0.0005 | 0.0004 | 0.0004 | 0.0015 | 0.0014 | 0.0008 | 0.0012 |
| VocabExt$_{RND}$ | 0.0002 | 0.0001 | 0.0002 | 0.0000 | 0.0020 | 0.0017 | 0.0034 | 0.0022 | 0.0005 | 0.0006 | 0.0006 | 0.0006 | 0.0006 | 0.0002 | 0.0015 | 0.0001 |
| LC-Rec | 0.0021 | 0.0014 | 0.0027 | 0.0016 | 0.0010 | 0.0007 | 0.0006 | 0.0006 | 0.0010 | 0.0009 | 0.0015 | 0.0010 | 0.0014 | 0.0019 | 0.0013 | 0.0019 |
| Mender$_{Emb}$ | 0.0011 | 0.0005 | 0.0017 | 0.0007 | 0.0007 | 0.0007 | 0.0017 | 0.0010 | 0.0015 | 0.0010 | 0.0023 | 0.0012 | 0.0035 | 0.0030 | 0.0040 | 0.0031 |
| Mender$_{Tok}$ | 0.0007 | 0.0005 | 0.0005 | 0.0004 | 0.0004 | 0.0001 | 0.0012 | 0.0002 | 0.0019 | 0.0011 | 0.0022 | 0.0012 | 0.0006 | 0.0004 | 0.0004 | 0.0003 |
| *Fine-grained steering* | | | | | | | | | | | | | | | | |
| TIGER | 0.0006 | 0.0004 | 0.0006 | 0.0004 | 0.0040 | 0.0024 | 0.0065 | 0.0032 | 0.0010 | 0.0006 | 0.0032 | 0.0011 | 0.0005 | 0.0003 | 0.0010 | 0.0004 |
| VocabExt$_{RND}$ | 0.0007 | 0.0005 | 0.0006 | 0.0005 | 0.0005 | 0.0004 | 0.0019 | 0.0009 | 0.0009 | 0.0004 | 0.0010 | 0.0004 | 0.0010 | 0.0005 | 0.0011 | 0.0004 |
| LC-Rec | 0.0034 | 0.0022 | 0.0054 | 0.0028 | 0.0009 | 0.0004 | 0.0018 | 0.0007 | 0.0016 | 0.0010 | 0.0024 | 0.0012 | 0.0014 | 0.0006 | 0.0020 | 0.0007 |
| Mender$_{Emb}$ | 0.0009 | 0.0005 | 0.0013 | 0.0007 | 0.0017 | 0.0013 | 0.0015 | 0.0012 | 0.0020 | 0.0017 | 0.0015 | 0.0015 | 0.0024 | 0.0014 | 0.0039 | 0.0019 |
| Mender$_{Tok}$ | 0.0004 | 0.0000 | 0.0010 | 0.0003 | 0.0012 | 0.0007 | 0.0010 | 0.0006 | 0.0008 | 0.0004 | 0.0010 | 0.0004 | 0.0005 | 0.0003 | 0.0004 | 0.0003 |
| *Coarse-grained steering* | | | | | | | | | | | | | | | | |
| TIGER | 0.0000 | 0.0000 | 0.0000 | 0.0000 | 0.0001 | 0.0000 | 0.0001 | 0.0001 | 0.0001 | 0.0001 | 0.0001 | 0.0001 | 0.0001 | 0.0001 | 0.0002 | 0.0001 |
| VocabExt$_{RND}$ | 0.0001 | 0.0000 | 0.0001 | 0.0000 | 0.0003 | 0.0002 | 0.0002 | 0.0000 | 0.0004 | 0.0003 | 0.0002 | 0.0002 | 0.0002 | 0.0001 | 0.0004 | 0.0001 |
| LC-Rec | 0.0005 | 0.0003 | 0.0008 | 0.0004 | 0.0006 | 0.0003 | 0.0012 | 0.0005 | 0.0007 | 0.0005 | 0.0009 | 0.0005 | 0.0005 | 0.0004 | 0.0008 | 0.0004 |
| Mender$_{Emb}$ | 0.0000 | 0.0000 | 0.0004 | 0.0001 | 0.0008 | 0.0005 | 0.0000 | 0.0002 | 0.0009 | 0.0006 | 0.0009 | 0.0005 | 0.0005 | 0.0002 | 0.0010 | 0.0003 |
| Mender$_{Tok}$ | 0.0002 | 0.0001 | 0.0005 | 0.0002 | 0.0015 | 0.0011 | 0.0017 | 0.0011 | 0.0003 | 0.0002 | 0.0009 | 0.0004 | 0.0005 | 0.0003 | 0.0002 | 0.0001 |
| *Sentiment following* | | | | | | | | | | | | | | | | |
| TIGER | 0.0000 | - | 0.0000 | - | 0.0000 | - | 0.0000 | - | 0.0000 | - | 0.0000 | - | 0.0000 | - | 0.0000 | - |
| VocabExt$_{RND}$ | 0.0000 | - | 0.0000 | - | 0.0012 | - | 0.0005 | - | 0.0000 | - | 0.0000 | - | 0.0029 | - | 0.0010 | - |
| LC-Rec | 0.0003 | - | 0.0007 | - | 0.0006 | - | 0.0012 | - | 0.0003 | - | 0.0007 | - | 0.0016 | - | 0.0014 | - |
| Mender$_{Emb}$ | 0.0001 | - | 0.0001 | - | 0.0003 | - | 0.0007 | - | 0.0002 | - | 0.0005 | - | 0.0003 | - | 0.0020 | - |
| Mender$_{Tok}$ | 0.0011 | - | 0.0012 | - | 0.0014 | - | 0.0003 | - | 0.0000 | - | 0.0002 | - | 0.0012 | - | 0.0014 | - |
| *History consolidation* | | | | | | | | | | | | | | | | |
| TIGER | 0.0000 | 0.0000 | 0.0000 | 0.0000 | 0.0000 | 0.0000 | 0.0000 | 0.0000 | 0.0000 | 0.0000 | 0.0000 | 0.0000 | 0.0000 | 0.0000 | 0.0000 | 0.0000 |
| VocabExt$_{RND}$ | 0.0001 | 0.0001 | 0.0007 | 0.0003 | 0.0017 | 0.0016 | 0.0020 | 0.0017 | 0.0009 | 0.0008 | 0.0006 | 0.0007 | 0.0023 | 0.0027 | 0.0028 | 0.0028 |
| LC-Rec | 0.0009 | 0.0006 | 0.0012 | 0.0007 | 0.0012 | 0.0007 | 0.0012 | 0.0007 | 0.0008 | 0.0003 | 0.0018 | 0.0007 | 0.0014 | 0.0019 | 0.0012 | 0.0018 |
| Mender$_{Emb}$ | 0.0011 | 0.0005 | 0.0018 | 0.0007 | 0.0007 | 0.0003 | 0.0005 | 0.0002 | 0.0006 | 0.0008 | 0.0015 | 0.0007 | 0.0003 | 0.0007 | 0.0006 | 0.0008 |
| Mender$_{Tok}$ | 0.0008 | 0.0006 | 0.0007 | 0.0006 | 0.0005 | 0.0000 | 0.0005 | 0.0001 | 0.0015 | 0.0013 | 0.0014 | 0.0013 | 0.0030 | 0.0023 | 0.0038 | 0.0025 |

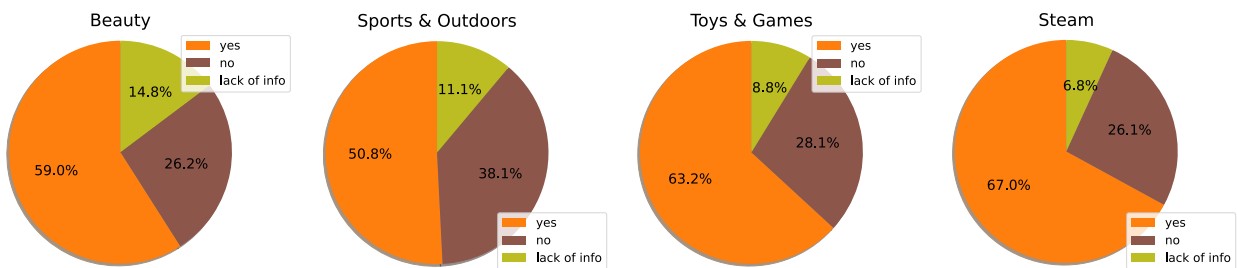

Figure 18: Manual inspection results for the question 'Given that a user preference accurately approximates the user's preferences, is it related to the target item?" for the four different datasets used for approximating user preferences.

Table 7: Performance, training time and inference time on an A100 GPU for $\text{Mender}_{\text{Emb}}$, $\text{Mender}_{\text{Tok}}$, and traditional sequential recommendation system SASRec (Kang & McAuley, 2018) on Beauty, Sports and Outdoors, Toys and Games, and Steam.

| Method | Dataset | Train time | Inference time | NDGC@10 | Recall@10 |
|---|---|---|---|---|---|
| SASRec | Beauty | 293min | 8ms | $0.0227 \pm 0.0004$ | $0.0528 \pm 0.0006$ |
| | Sports and Outdoors | 447min | 9ms | $0.0118 \pm 0.0002$ | $0.0271 \pm 0.0005$ |
| | Toys and Games | 280min | 5ms | $0.0267 \pm 0.0002$ | $0.0615 \pm 0.0002$ |
| | Steam | 280min | 5ms | $0.1469 \pm 0.0002$ | $0.1781 \pm 0.0004$ |
| $\text{Mender}_{\text{Emb}}$ | Beauty | 127min | 453ms | $0.0405 \pm 0.001$ | $0.0755 \pm 0.0017$ |
| | Sports and Outdoors | 374min | 194ms | $0.0215 \pm 0.0007$ | $0.0394 \pm 0.0017$ |
| | Toys and Games | 239min | 178ms | $0.0342 \pm 0.0015$ | $0.0653 \pm 0.0015$ |
| | Steam | 231min | 179ms | $0.123 \pm 0.0031$ | $0.182 \pm 0.004$ |
| $\text{Mender}_{\text{Tok}}$ | Beauty | 2324min | 562ms | $0.0508 \pm 0.0002$ | $0.0937 \pm 0.0012$ |
| | Sports and Outdoors | 2350min | 210ms | $0.0234 \pm 0.0004$ | $0.0427 \pm 0.0005$ |
| | Toys and Games | 1021min | 227ms | $0.0432 \pm 0.0012$ | $0.0799 \pm 0.0022$ |
| | Steam | 2330min | 222ms | $0.156 \pm 0.0003$ | $0.204 \pm 0.0004$ |

Table 8: Performance of $\text{Mender}_{\text{Tok}}$ when being trained on the single matched preference compared to training on all five generated user preferences on the Amazon datasets. For sentiment following we report $m@10$ instead of Recall@10.

| Methods | Beauty | | Sports | | Toys | |
|---|---|---|---|---|---|---|
| | Recall @10 | NDCG @10 | Recall @10 | NDCG @10 | Recall @10 | NDCG @10 |
| Recommendation | | | | | | |
| $\text{Mender}_{\text{Tok}}$ | 0.0937 | 0.0508 | 0.0427 | 0.0234 | 0.0799 | 0.0432 |
| $\text{Mender}_{\text{Tok}}$-AllPrefs | 0.0131 | 0.0066 | 0.0063 | 0.0037 | 0.0074 | 0.0039 |
| Fine-grained steering | | | | | | |
| $\text{Mender}_{\text{Tok}}$ | 0.0844 | 0.0444 | 0.0324 | 0.0159 | 0.0639 | 0.0321 |
| $\text{Mender}_{\text{Tok}}$-AllPrefs | 0.0014 | 0.0006 | 0.0009 | 0.0004 | 0.0018 | 0.0009 |
| Coarse-grained steering | | | | | | |
| $\text{Mender}_{\text{Tok}}$ | 0.0161 | 0.0080 | 0.0045 | 0.0021 | 0.0060 | 0.0029 |
| $\text{Mender}_{\text{Tok}}$-AllPrefs | 0.0006 | 0.0002 | 0.0003 | 0.0002 | 0.0006 | 0.0003 |
| Sentiment following | | | | | | |
| $\text{Mender}_{\text{Tok}}$ | 0.0053 | - | 0.0042 | - | 0.0017 | - |
| $\text{Mender}_{\text{Tok}}$-AllPrefs | 0.0008 | - | 0.0001 | - | 0.0005 | - |
| History consolidation | | | | | | |
| $\text{Mender}_{\text{Tok}}$ | 0.0720 | 0.0388 | 0.0345 | 0.0187 | 0.0700 | 0.0377 |
| $\text{Mender}_{\text{Tok}}$-AllPrefs | 0.0089 | 0.0041 | 0.0063 | 0.0038 | 0.0046 | 0.0025 |

Table 9: Performance for Mender$_{\text{Tok}}$ compared to Mender$_{\text{Tok}}$-XXL for all datasets trained on recommendation data. We report average performance across three random seeds. For sentiment following we report $m@k$ for $k \in \{5, 10\}$ instead of Recall@k.

| Methods | Sports and Outdoors | | | | Beauty | | | | Toys and Games | | | | Steam | | | |
|---|---|---|---|---|---|---|---|---|---|---|---|---|---|---|---|---|
| | Recall@5 | NDCG@5 | Recall@10 | NDCG@10 | Recall@5 | NDCG@5 | Recall@10 | NDCG@10 | Recall@5 | NDCG@5 | Recall@10 | NDCG@10 | Recall@5 | NDCG@5 | Recall@10 | NDCG@10 |
| **Recommendation** | | | | | | | | | | | | | | | | |
| Mender$_{\text{Tok}}$ | 0.0282 | 0.0188 | 0.0427 | 0.0234 | 0.0605 | 0.0401 | 0.0937 | 0.0508 | 0.0533 | 0.0346 | 0.0799 | 0.0432 | 0.1682 | 0.1441 | 0.2037 | 0.1555 |
| Mender$_{\text{Tok}}$-XXL | 0.0302 | 0.0201 | 0.0443 | 0.0247 | 0.0523 | 0.0341 | 0.0802 | 0.0431 | 0.0466 | 0.0307 | 0.0691 | 0.0380 | 0.1702 | 0.1472 | 0.2033 | 0.1579 |
| **Fine-grained steering** | | | | | | | | | | | | | | | | |
| Mender$_{\text{Tok}}$ | 0.0190 | 0.0116 | 0.0324 | 0.0159 | 0.0534 | 0.0344 | 0.0844 | 0.0444 | 0.0378 | 0.0237 | 0.0639 | 0.0321 | 0.0218 | 0.0137 | 0.0357 | 0.0182 |
| Mender$_{\text{Tok}}$-XXL | 0.0338 | 0.0206 | 0.0551 | 0.0274 | 0.0495 | 0.0319 | 0.0787 | 0.0412 | 0.0423 | 0.0264 | 0.0681 | 0.0347 | 0.0246 | 0.0157 | 0.0394 | 0.0204 |
| **Coarse-grained steering** | | | | | | | | | | | | | | | | |
| Mender$_{\text{Tok}}$ | 0.0023 | 0.0013 | 0.0045 | 0.0021 | 0.0094 | 0.0059 | 0.0161 | 0.0080 | 0.0032 | 0.0020 | 0.0060 | 0.0029 | 0.0045 | 0.0028 | 0.0085 | 0.0041 |
| Mender$_{\text{Tok}}$-XXL | 0.0096 | 0.0058 | 0.0172 | 0.0082 | 0.0104 | 0.0062 | 0.0184 | 0.0087 | 0.0086 | 0.0053 | 0.0140 | 0.0070 | 0.0048 | 0.0029 | 0.0091 | 0.0043 |
| **Sentiment following** | | | | | | | | | | | | | | | | |
| Mender$_{\text{Tok}}$ | 0.0035 | - | 0.0042 | - | 0.0043 | - | 0.0053 | - | 0.0016 | - | 0.0017 | - | 0.0084 | - | 0.0110 | |
| Mender$_{\text{Tok}}$-XXL | 0.0044 | - | 0.0064 | - | 0.0076 | - | 0.0103 | - | 0.0020 | - | 0.0048 | - | 0.0135 | - | 0.0197 | |
| **History consolidation** | | | | | | | | | | | | | | | | |
| Mender$_{\text{Tok}}$ | 0.0234 | 0.0151 | 0.0345 | 0.0187 | 0.0457 | 0.0304 | 0.0720 | 0.0388 | 0.0467 | 0.0302 | 0.0700 | 0.0377 | 0.0490 | 0.0317 | 0.0745 | 0.0399 |
| Mender$_{\text{Tok}}$-XXL | 0.0223 | 0.0144 | 0.0334 | 0.0180 | 0.0362 | 0.0235 | 0.0574 | 0.0303 | 0.0383 | 0.0253 | 0.0582 | 0.0317 | 0.1225 | 0.1015 | 0.1458 | 0.1091 |

