# OpenReview forum: "Preference Discerning with LLM-Enhanced Generative Retrieval"
_TMLR — Accepted by TMLR_

### Review · Reviewer_uuYr · 2025-03-14

**Summary Of Contributions:**

This paper studies the problem of sequential generative recommendation. The authors propose a new paradigm called preference discerning, where the recommendation model explicitly conditions on user preferences in natural language. They further provide a holistic evaluation across various scenarios, including preference steering and sentiment following. The effectiveness of the proposed method is demonstrated on several benchmark datasets.

**Audience:**

Yes

**Claims And Evidence:**

Yes

**Requested Changes:**

1. Make Algorithm 1 clearer. Some parameters, such as t, are not defined.
2. The figure 4 results are very hard to read. I would recommend changing it into a Table. There is also a lack of subfigure captions to distinguish between subdatasets.

**Strengths And Weaknesses:**

Strength
- The paper is overall clearly written and easy to read.
- The proposed method is demonstrated to be effective on many datasets.
- Extensive model studies are conducted.


Weakness
- Some of the model design is not clearly mentioned. For example, from Section 3.2, it seems that the author is making a multimodal recommendation. However, there is no explicit explanation of what multimodal large language models are adopted as the backbone.
- Some illustrations are not clear, e.g., in Algorithm 1, many parameters are not defined.
- The presentation of the results is hard to read.

---

> ### Author Response · Authors · 2025-03-21
>
> Dear reviewer uuYr,
>
> Thank you for the constructive feedback and the suggested changes to improve our manuscript.
>
> We have made the following changes to accommodate the requests of the reviewer, all of which are highlighted in **teal** in the revised version:
>
> - We added a definition for $T_u$ in Algorithm 1 for clarification, all symbols should be defined now
> - We explicitly mention the choice of language encoder in Section 3.3 and clarify that we do not use a multimodal encoder, but the multimodality stems from semantic IDs being a different modality than natural language
> - We added subcaptions in Figure 4 to clarify what results are on which datasets along with a brief interpretation of results.
> - The results in figure 4 are also visible in Table 1 in the row “Recommendation” in the  “Recall@10 column” for the different datasets. We clarified this in the text.
>
> If there are any remaining requests, we will gladly incorporate those into the manuscript.

---

> > ### Author Response · Authors · 2025-04-11
> >
> > Dear Reviewer uuYr,
> >
> > We would like to kindly inquire whether our response and revision sufficiently addressed your concerns. We would greatly appreciate your feedback which substantially contributes to improving our manuscript.

---

> ### Author Response · Authors · 2025-05-07
> **Friendly Reminder: TMLR Submission Review Status**
>
> Dear Reviewer uuYr,
>
> We wanted to kindly inquire if you have had a chance to review the changes made in response to your feedback.
> Your confirmation on whether our revisions addressed your concerns would be greatly appreciated.
>
> Thank you for your time and consideration!

---

> ### Comment · Reviewer_uuYr · 2025-05-19
>
> I would like to thank the author for the thorough reply, which addresses most of my concerns. I have no additional feedback.

---

### Review · Reviewer_Kc2f · 2025-03-17

**Summary Of Contributions:**

This paper introduces "preference discerning" as a novel paradigm for sequential recommendation systems. I found their approach interesting - they use a two-stage process where they first extract user preferences using LLMs, then explicitly condition a recommendation model on these preferences during inference.

The main contributions as I see them are:

They propose a way to dynamically adapt recommendations based on changing user preferences without retraining models, which addresses a practical limitation of current systems.
They've developed a benchmark that evaluates preference discerning across five different aspects: preference-based recommendation, sentiment following, fine/coarse-grained steering, and history consolidation.
They introduce Mender (and its variants MenderTok/MenderEmb), which combines pre-trained language encoders with generative retrieval. The architecture uses cross-attention to fuse language understanding with collaborative filtering.
Their experiments on Amazon reviews and Steam datasets show meaningful improvements over existing methods, particularly in preference-based recommendation where they achieve up to 45% relative improvement.
The core idea - that we can represent user preferences in natural language and use them to steer recommendations - strikes me as a promising direction, especially since this provides a more interpretable interface for recommendation systems.

**Audience:**

No

**Claims And Evidence:**

Yes

**Requested Changes:**

Rewrite the introduction: The paper desperately needs a clearer problem statement before diving into the solution. Why should I care about preference discerning? What specific limitations does it address? Start with the problem, then introduce your approach.
Add background section: Before Section 3, include background on sequential recommendation, generative retrieval, and why existing approaches struggle with adapting to user preferences. This would help readers understand where your work fits.
Clarify preference generation: Provide concrete examples of how preferences are extracted from user data. Show actual prompts used with the LLM and example outputs for a few representative cases. This is crucial for reproducibility.
Compare with non-generative methods: Include at least a brief comparison with traditional recommendation approaches to situate your work in the broader landscape. SASRec appears in an appendix table but deserves discussion in the main text.

Recommended Changes:
Better discuss limitations: The limitations section (Section 5) feels like an afterthought. Expand on potential issues with privacy, generalizability, and computational requirements.
Include worked examples: A concrete walk-through of how a specific user's preferences are extracted and used would make your approach much more tangible.
Improve figures: Figure 4 packs in too much information without enough explanation. Consider breaking this down or adding annotations to help readers interpret the results.
Address computational costs: Discuss the practical implications of using LLMs for preference extraction. Is this feasible at scale? What are the trade-offs?
Consolidate implementation details: Pull important architecture details from the appendices into the main text so readers don't have to jump back and forth.
Add failure analysis: Where does your approach struggle? Are there particular types of preferences or domains where it doesn't work well?
Clarify training vs. inference use: I got confused about whether user preferences are used just during training or also at inference time. Make this distinction clearer throughout.

**Strengths And Weaknesses:**

Strengths:
Practical relevance: I appreciate that the authors tackle a real problem - current recommendation models can't adapt to changing user preferences without costly retraining.
Comprehensive evaluation: The benchmark they created tests multiple capabilities rather than just focusing on standard recommendation metrics. This gives a much better picture of how these models actually behave.
Solid empirical results: Mender clearly outperforms baselines on most evaluation axes. The improvements look substantial enough to matter in practice.
Ablation studies: Fig. 5 and the associated discussion were particularly useful - I liked seeing how different training data combinations affect the model's capabilities.
Manual verification: Their human evaluation of preference quality (Appendix F) strengthens confidence in the approach - finding that ~75% of generated preferences correctly reflect user intentions is important validation.
Weaknesses:
Confusing introduction: I had to re-read the first two pages several times to understand what problem they're solving. The authors jump into their solution without properly setting up the problem first.
Missing background: Section 3 drops me right into the methodology without any context. As someone not deeply familiar with generative retrieval, I struggled to follow their technical approach without more background.
Vague preference generation process: Algorithm 1 doesn't tell me much about how preferences are actually generated. What prompt do they use? How do they ensure quality? The explanation is too thin.
Limited dataset scope: I wonder how well this would work beyond e-commerce. Would it generalize to music, video, or news recommendation where user preferences might be more abstract?
Efficiency concerns: While they mention MenderEmb is 5x faster than MenderTok, there's little discussion about the computational cost of their approach. Using LLMs for preference extraction must have significant overhead - is this practical?
Missing comparisons: They focus almost exclusively on comparing against generative retrieval methods. I would've liked to see how this compares to traditional recommendation approaches too.
Implementation details buried: Important details about model architecture are scattered between the main text and appendices, making it hard to get a complete picture of their approach.

---

> ### Author Response · Authors · 2025-03-21
>
> Der reviewer Kc2f,
>
> Thank you for the constructive feedback and the suggested changes to substantially improve the clarity of our manuscript.
> We have made the following revisions, which are highlighted in **brown** in the re-uploaded version:
>
> - **Introduction:**  We revised the motivation and parts of the abstract to make our motivation more explicit.
> - **Background section:** We added Section 3.1 which provides additional background on sequential recommendation, traditional approaches and generative retrieval.
> - **Comparison to traditional methods:** We provide a brief comparison for recommendation to traditional methods, FDSA, SASRec, S3-Rec, BERT4Rec, GRU4Rec, HGN, etc. in Table 2 in Appendix A3 and briefly added an interpretation of these results in the main text in Section 4.2. Our TIGER implementation outperforms most of the traditional methods, and MenderTok significantly improves upon TIGER.
> - **Prompt examples:** We provide examples for the prompts we used in Appendix C, we added an additional reference in the main text, pointing to the respective section.
> - **Implementation details:** We provide the most important implementation details in the first two paragraphs in Section 4. We provide implementation details for training of the semantic IDs and the Transformer in Appendix A1, and A2, respectively. Since there are many technical details and our submission is limited by page number, we cannot pull all of them into the main text. To further facilitate reproducibility, we are in the process of open sourcing our code, which contains all necessary configurations to reproduce all our results.
> - **Clarification:** We adapted the figures with subcaptions and added a brief interpretation of the take-aways for Figures 4 and 5 in the main caption.
> - **Clarification:** We clarify in Section 4 in the second paragraph that we use user preferences for both training and inference
> - **Failure Cases**: We discuss in Section 4.2 that Mender’s steering capabilities are limited on the Steam benchmark unlike for Amazon datasets. We hypothesize this is due to the difference in item distributions, which is very different for Steam, as we show in Figure 9 and elaborate on in Section 4.2. Generally, the performance of Mender relies heavily on high-quality user preferences, therefore manual inspection of preferences is crucial. Potential failure cases can arise if generated user preferences are too generic and do not contain information relevant to items. In this case we would expect Mender to perform equivalent to the TIGER baseline. We now also explicitly mention this in our revised Limitation section.
> - **Limitations:** We revised the Limitations section to include discussion on generalization, computational complexity and scaling of the preference approximation pipeline
>
> If there are any remaining requests, we will gladly incorporate those into the manuscript.

---

> > ### Author Response · Authors · 2025-04-11
> >
> > Dear Reviewer Kc2f,
> >
> > We would like to kindly inquire whether our response and revision sufficiently addressed your concerns. We would greatly appreciate your feedback which substantially contributes to improving our manuscript.

---

> > > ### Comment · Reviewer_Kc2f · 2025-04-18
> > > **Official Comment**
> > >
> > > The author has addressed my concerns very well, therefore I have no further comments. I appreciate the author's efforts and dedication

---

> ### Author Response · Authors · 2025-04-18
>
> Dear reviewer KC2f,
>
> We would like to thank you for the positive response and are glad we could address all of your concerns. Thank you for your time and effort invested in reviewing our manuscript. Your feedback has been invaluable and led to substantial improvements of our manuscript.
>
> Thank you!

---

### Review · Reviewer_R3DY · 2025-04-13

**Summary Of Contributions:**

This paper proposes a new preference recognition paradigm, which uses LLM to summarize the user's historical evaluation and historical interactive item information into user preferences, and then guides the multimodal recommendation model. Based on this paradigm, this paper proposes five subtasks to measure the ability of multimodal recommendation models to recognize user preferences. Then, it proposes a new approach, mender, which is a multimodal model integrating semantic id and language preferences, and achieves the most advanced performance in benchmarking.

**Audience:**

Yes

**Broader Impact Concerns:**

This work has no moral or ethical concerns.

**Claims And Evidence:**

Yes

**Requested Changes:**

Please refer to “Strengths And Weaknesses”.

**Strengths And Weaknesses:**

Strengths:
1. This paper discusses the important issue of how the recommendation system can obtain user preferences more effectively. The new paradigm of preference recognition proposed in this paper has room for application.
2. The evaluation indicators proposed in this paper are detailed. These metrics take into account the varying degrees of change in user preferences, as well as the likes and dislikes of users.
3. The paper has carried out sufficiently detailed experiments. It has been tested on multiple datasets on Amazon and Steam and on multiple platforms to prove its effectiveness.
4. The part of the paper on how LLM can extract user preferences is valuable. This paper explains the influence of the thickness and granularity of extraction preferences on the experimental results, and answers the possible questions in the main text. In addition, the paper also explains how the method used to format the user evaluation, which can provide reference for the future work.

Weaknesses:
1. In section 3.5, when the authors propose new measures, they state that "Since $p\_u^{(t-1)}$ is generated based only on information about past items, there is no information leak...". This statement requires further checking. First, in the formula for calculating $p\_u^{(t-1)}$, $i\_u^{t}$is actually the input. Second, in practical application, it seems impractical to take $i_u^{t}$ as the input, so the rationality of this metric may require further discussion.
2. Further explanation and experiments are suggested to determine the positive effect of user preference information proposed in this paper. The effect of user preferences was mentioned in the smile experiment conducted in the paper, but from the experimental results, it seems that its effect alone is lower than the effect of object description alone.
3. The LLM used in this paper was slightly large (LLaMA-3-70B-Instruct) and may not be widely applied to the community. It is suggested that the results of experiments conducted on smaller models or distillation models can be supplemented.

---

> ### Author Response · Authors · 2025-04-14
>
> Dear reviewer R3DY:
>
> We would like to thank you for the constructive feedback and the raised concerns.
> We have made the following changes to accommodate the requests of the reviewer, all of which are highlighted in **red** in the revised version:
>
>  1. Thank you for pointing out this inconsistency. Indeed there has been a typo, specifically the user preference associated with item $i_u^t$ should be $p_u^t$ instead of $p_u^{t - 1}$. Furthermore, in Eq. (1) the preference $p_u^t$ is then obtained according to cosine similarity of $i_u^t$ with preferences $p \in P_u^{t-1}$, hence **the ground truth item is associated with preferences obtained from past items.** We have corrected this in the revised version, thank you again for pointing this out as this is indeed crucial.
>  2. The main focus of our paper is in developing a new paradigm called “preference discerning” to approximate user preferences from previous comments, reviews, or recent activity and to provide them to the recommendation model such that it can dynamically adapt its recommendations. We have shown in various experiments (such as Table1, Figures 4, 5, and 6) that in our holistic evaluation framework the capabilities of recommender systems not leveraging user preferences are very limited in dimensons such as history consolidation, fine- and coarse-grained steering, and sentiment following. Additionally, we have shown that, by using user preferences, the overall recommendation performance consistently improves (see TIGER vs Mender in Table 1, Figures 4 & 5. In Figure 6 we additionally demonstrated that user preferences are crucial for the superior performance of Mender and that item descriptors alone are insufficient. We hope this summary clarifies the importance of the preference approximation and conditioning. Finally, we also wanted to ask for clarification on the term “smile experiment” in the reviewer’s questions, as it is unclear to us what experiment this refers to.
>  3.  We leveraged the LLama-3-70B-Instruct model, as it has been widely used in the community, and most importantly, its weights are open sourced. This allows the community to build upon our work, and  encourages open-source progress in preference generation by future work, while being a very performant model resulting in high quality preference approximation.
>
>
> We thank the reviewer again for the insightful feedback and hope these clarifications have address your concern.

---

> > ### Comment · Reviewer_R3DY · 2025-04-18
> >
> > Thanks! The author's response comprehensively answered the raised questions: corrected the inconsistencies of key formulas and clarified the logic of preference derivation through the revised paper; The principle of model selection (llama-3-70b - directive) balances performance with open-source repeatability. For me, there are no major issues for publication consideration.

---

> ### Author Response · Authors · 2025-04-18
>
> Dear reviewer R3DY,
>
> We would like to thank you for the positive response and your time and effort invested in reviewing our manuscript. Your feedback has been invaluable and led to substantial improvements of our manuscript.
>
> Thank you!

---

> > ### Comment · Reviewer_R3DY · 2025-05-21
> >
> > I would like to thank the author for their reply.  My concerns have been addressed.

---

### Decision · Action_Editor_Jk5V · 2025-06-27

**Recommendation:** Accept with minor revision

**Audience:**

Yes

**Audience Explanation:**

This paper introduces a new paradigm for preference-aware multimodal recommendation and provides solid empirical results supported by a comprehensive benchmark. While the overall methodological novelty is moderate, the work is well-executed and the authors have been responsive in addressing reviewer concerns. The paper is sufficiently clear and relevant to the community. I recommend acceptance with minor revisions, contingent on the authors incorporating the revisions outlined in their rebuttal.

**Claims And Evidence:**

Yes

**Claims Explanation:**

The claims made in this paper are supported by empirical evaluations.